# An Exponential Improvement on the Memorization Capacity of Deep Threshold Networks

**Shashank Rajput**
University of Wisconsin - Madison
`rajput3@wisc.edu`

**Kartik Sreenivasan**
University of Wisconsin - Madison
`ksreenivasa2@wisc.edu`

**Dimitris Papailiopoulos**
University of Wisconsin - Madison
`dimitris@papail.io`

**Amin Karbasi**
Yale University
`amin.karbasi@yale.edu`

## Abstract

It is well known that modern deep neural networks are powerful enough to memorize datasets even when the labels have been randomized. Recently, Vershynin (2020) settled a long standing question by Baum (1988), proving that *deep threshold* networks can memorize $n$ points in $d$ dimensions using $\widetilde{\mathcal{O}}(e^{1/\delta^2} + \sqrt{n})$ neurons and $\widetilde{\mathcal{O}}(e^{1/\delta^2}(d + \sqrt{n}) + n)$ weights, where $\delta$ is the minimum distance between the points. In this work, we improve the dependence on $\delta$ from exponential to almost linear, proving that $\widetilde{\mathcal{O}}(\frac{1}{\delta} + \sqrt{n})$ neurons and $\widetilde{\mathcal{O}}(\frac{d}{\delta} + n)$ weights are sufficient. Our construction uses Gaussian random weights only in the first layer, while all the subsequent layers use binary or integer weights. We also prove new lower bounds by connecting memorization in neural networks to the purely geometric problem of separating $n$ points on a sphere using hyperplanes.

## 1  Introduction

The current paradigm of training neural networks is to train the networks until they fit the training dataset perfectly (Zhang et al., 2017; Arpit et al., 2017). This means that the network is able to output the exact label for every sample in the training set, a phenomenon known as *interpolation*. Quite interestingly, modern deep networks have been known to be powerful enough to interpolate even randomized labels (Zhang et al., 2017; Liu et al., 2020), a phenomenon that is usually referred to as *memorization* (Yun et al., 2019; Vershynin, 2020; Bubeck et al., 2020), where the networks can interpolate any arbitrary labeling of the dataset.

Given that memorization is a common phenomenon in modern deep learning, a reasonable question to ask is that of how big a neural network needs to be so that it can memorize a dataset of $n$ points in $d$ dimensions. This question has been of interest since the 60s (Cover, 1965; Baum, 1988; Mitchison & Durbin, 1989; Sontag, 1990; Huang et al., 1991; Sartori & Antsaklis, 1991). In particular, Baum (1988) proved that a single hidden layer threshold network with $\mathcal{O}(\max(n, d))$ weights and $\mathcal{O}(\lceil n/d \rceil)$ neurons can memorize any set of $n$ points in $d$ dimensions, as long as they are in general position.[1] Baum asked if going deeper could increase the memorization power of threshold networks, and in particular if one could reduce the number of neurons needed for memorization to $\mathcal{O}(\sqrt{n})$. While for ReLU activated networks, Yun et al. (2019) were able to prove that indeed only $\mathcal{O}(\sqrt{n})$ neurons were sufficient for memorization using deeper networks, the same question remained open for threshold networks.

---

[1]The "general position" assumption of $n$ vectors in $\mathbb{R}^d$ means that any collection of $d$ of these $n$ vectors are linearly independent.

| Reference | # Neurons | # Weights | Assumptions |
|-----------|-----------|-----------|-------------|
| Baum (1988) | $\mathcal{O}(\lceil n/d \rceil)$ | $\mathcal{O}(\max(n,d))$ | General Position |
| Huang et al. (1991) | $\mathcal{O}(n)$ | $\mathcal{O}(nd)$ | None |
| Vershynin (2020) | $\mathcal{O}(e^{1/\delta^2}\log^p(n) + \sqrt{n}\log^{2.5}(n))$ | $\mathcal{O}(e^{1/\delta^2}(d+\sqrt{n}) + n\log^5(n))$ | $\delta$-separated, points lie on the unit sphere |
| **Ours, Theorem 1** | $\mathcal{O}\left(\dfrac{\log^2 n}{\delta} + \sqrt{n}\log^2 n\right)$ | $\mathcal{O}\left(\dfrac{(d+\log n)\log n}{\delta} + n\log^2 n\right)$ | $\delta$-separated |

Table 1: Table comparing the upper bounds for the parameters needed for a threshold network to memorize a dataset. General position assumption in $\mathbb{R}^d$ means that no more than $d$ points lie on a $d-1$ dimensional hyperplane. Note: $\log^p(n)$ denotes a poly-log factor in $n$.

Recently, Vershynin (2020) was able to answer Baum's question in the positive by proving that indeed $\widetilde{\mathcal{O}}(e^{1/\delta^2} + \sqrt{n})$ neurons were sufficient to memorize a set of $n$ points on a unit sphere separated by a distance at least $\delta$. Many recent works study the memorization power of neural networks under separation assumptions, *e.g.,* Bubeck et al. (2020); Vershynin (2020); Park et al. (2020). One reason why before Vershynin's work, it was unclear whether going deeper was helpful for threshold networks was that unlike ReLU or sigmoid functions, the threshold activation function, *i.e.,* $\sigma(z) = \mathbf{1}_{z \geq 0}$, prohibits neurons from passing 'amplitude information' to the next layer.

Although the dependence of $O(\sqrt{n})$ on the number of neurons was achieved by the work of Vershynin, it is unclear that the exponential dependence on distance and the requirement that the points lie on a sphere is fundamental. In this work, we lift the spherical requirement and offer an exponential improvement on the dependence on $\delta$.

**Theorem 1.** *(Informal) There exists a threshold neural network with* $\widetilde{\mathcal{O}}\left(\frac{1}{\delta} + \sqrt{n}\right)$ *neurons and* $\widetilde{\mathcal{O}}\left(\frac{d}{\delta} + n\right)$ *weights that can memorize any $\delta$-separated dataset of size $n$ in $d$ dimensions.*

Please see Definition 2 for the formal definition of $\delta$-separation; informally, it means that all the points have bounded norm and a distance of at least $\delta$ between them. Comparing the theorem above with Vershynin's result, we see that firstly we have reduced the dependence on $\delta$ from exponential to nearly linear; and secondly, in our upper bound for the number of weights, the $\delta$ and $n$ terms appear in summation rather than product (ignoring the logarithmic factors). Further, note that Vershynin (2020) needs the points to lie on a sphere, whereas we only need them to have a bounded $\ell^2$ norm. We have compared our results with the existing work in Table 1.

Our construction has $\mathcal{O}(\log \frac{\log n}{\delta})$ layers, of which only the first layer in our construction has real weights, which are i.i.d Gaussian. This layer has $\mathcal{O}(\frac{\log n}{\delta})$ neurons and $\mathcal{O}(\frac{d \log n}{\delta})$ weights. All the other layers have integer weights which are bounded and are on the of order $\mathcal{O}(\log n)$.

Baum (1988) also proved that there exists a dataset of $n$ points such that any threshold neural network would need at least $\lceil n/d \rceil$ neurons in the first layer to memorize it. However, the distance for this dataset is $\mathcal{O}(1/n)$.[2] Later, Baum & Haussler (1989) proved VC-dimension bounds which imply that $\min\{\sqrt{n}, n/d\}$ neurons are necessary in a network to memorize a dataset of $n$ points in $d$ dimensions, and this bound is independent of the minimum distance $\delta$. Our upper bound (Theorem 1) shows that if $\delta = \Omega(1/n)$, then we can memorize with only $\mathcal{O}(\frac{\log n}{\delta})$ neurons in the first layer. To complement our upper bound, we introduce a new, $\delta$-dependent lower bound below:

**Theorem 2.** *(Informal) There exists a $\delta$-separated dataset of size $n \in \left[\frac{d^2}{\delta}, \left(\frac{1}{\delta}\right)^{\frac{d}{2}}\right]$ such that any threshold network that can memorize it needs* $\widetilde{\Omega}\left(\frac{1}{\sqrt{\delta}}\right)$ *neurons in the first layer.*

The rest of the paper is divided into 7 sections. In Section 2, we provide the related works for memorization using neural networks. We provide definitions and notation in Section 3 and our main results in Section 4. Then, we briefly explain the constructions of our upper bound in Section 5.

---

[2]Here, we have rescaled the dataset to have maximum norm of any sample to be 1. This is done to make the dataset consistent with our minimum distance definition (see Assumption 2). The assumption in Park et al. (2020) also uses this kind of normalization.

Before exploring lower bounds for threshold networks, we provide sharp bounds on the minimum parameterization needed for any model (not necessarily a neural network) to memorize a $\delta$-separated dataset, in Section 6. We discuss our lower bound for threshold networks in Section 7. Finally, we conclude our paper with a brief conclusion in Section 8.

## 2 Related works

**Memorization with threshold activation.** There is a rich history of research into the memorization capacity of threshold networks. Baum (1988) showed that a single hidden layer network with $\mathcal{O}(\max(n, d))$ weights and $\mathcal{O}(\lceil n/d \rceil)$ neurons are sufficient to memorize a dataset, *if* the samples are in general positions. Baum also provided an instance of a dataset, where any threshold based network aiming to memorize the dataset would need at least $\lceil n/d \rceil$ neurons in the first layer. However, the construction was such that $\delta$ scaled inversely with $n$. In this work, we show that for any minimum distance $0 < \delta \le \frac{1}{2}$, the number of neurons in the first layer can be much smaller. Baum also proved, using the Function Counting Theorem (Cover, 1965), that $\mathcal{O}(n/\log n)$ connections are needed in any neural network that aims to memorize points in general position, regardless of the minimum distance.

Huang et al. (1991) and Sartori & Antsaklis (1991) proved that threshold networks could memorize any arbitrary set of points, that is, the points need not be in general position. However, these results needed $(n-1)d$ weights and $n-1$ neurons in the networks for memorization. Mitchison & Durbin (1989) and Kowalczyk (1997) provided upper and lower bounds when the requirement of memorizing the dataset exactly was relaxed to allow a recall error of up $50\%$. Sontag (1990) provided evidence that suggested that adding skip connections can double the memorization capacity of single hidden layer networks. Kotsovsky et al. (2020) extended Baum's result to multiclass classification setting, using bithresholding activation functions.

Recently, Vershynin (2020) was able to show that $\mathcal{O}(e^{1/\delta^2}(d + \sqrt{n}) + n)$ weights and $\mathcal{O}(e^{1/\delta^2} + \sqrt{n})$ neurons are sufficient to memorize a set of $n$ points on the unit sphere, separated by a distance of at least $\delta$. This answered the question by Baum (1988), which was whether the number of neurons could be reduced to $\mathcal{O}(\sqrt{n})$ by considering deeper networks. Note that Vershynin's and Baum's assumptions are fundamentally different and neither is stronger than the other. For example, Baum's construction would still need only $\mathcal{O}(\max(n, d))$ weights even if the points are infinitesimally close to each other, whereas Vershynin's construction would grow as $\delta$ decreases. On the other hand, Vershynin's construction works even if all the points lie on a low dimensional hyperplane, whereas Baum's general position assumption would get violated in this situation. Note that naïvely extending Baum's construction to the case where the points lie on a very low dimensional hyperplane would result in $\widetilde{\mathcal{O}}(nd)$ weights and $\widetilde{\mathcal{O}}(n)$ neurons[3], which is identical to the bounds by Huang et al. (1991).

**Memorization with ReLU and sigmoid activation.** Due to the popularity of the ReLU activation function, there has also been an interest in their memorization capacity. Zhang et al. (2017) proved that a single hidden layer ReLU network with $2n + d$ weights and $n$ neurons can memorize $n$ arbitrary points. Hardt & Ma (2017) proved that ReLU networks with residual connections also need only $\mathcal{O}(n)$ weights to memorize $n$ points. Yun et al. (2019) were able to prove that for ReLU, in fact, one needs only $\mathcal{O}(\sqrt{n})$ neurons to memorize $n$ points. Quite interestingly, the construction by Vershynin (2020) works for both threshold activated networks (as described above) as well as ReLU activated networks; and gives the same bounds for both architectures. Bubeck et al. (2020) show that under some minimum distance assumptions, ReLU networks can memorize (even real labels) with $\mathcal{O}(n/d)$ neurons such that the total weight (Bartlett, 1998) of the network is $\mathcal{O}(\sqrt{n})$, which is proved to be optimal for single hidden layer networks. Huang (2003) proved that a two-hidden layer network with $\mathcal{O}(\sqrt{n})$ sigmoid activated neurons suffices to memorize $n$ points. Recently, Park et al. (2020) have shown that ReLU or sigmoid activated networks with only $\mathcal{O}(n^{2/3} + \log(1/\delta))$ weights are sufficient to memorize $n$ points separated by a normalized distance of $\delta$.

**Over parameterization and generalization.** A large body of recent work attempts to explain why large neural networks, which are capable memorizers, can also generalize well, *e.g.,* Neyshabur et al. (2018, 2019); Belkin et al. (2020). In this work, we only study memorization, and omit any discussion about generalization. We do not believe our results explain anything about generalization.

---

[3]Please see Appendix A for a detailed explanation.

## 3 Preliminaries

In this work, we will consider feed forward neural networks $f : \mathbb{R}^d \to \{0, 1\}$ of the form

$$f(\boldsymbol{x}) := \sigma(b_L + \boldsymbol{w}_L^\top \sigma(\boldsymbol{b}_{L-1} + \boldsymbol{W}_{L-1}\sigma(\boldsymbol{b}_{L-2} + \boldsymbol{W}_{L-2}\sigma(\ldots \sigma(\boldsymbol{b}_1 + \boldsymbol{W}_1\boldsymbol{x})\ldots)))),$$

where $\sigma(\cdot)$ is the threshold activation. The network has $L$ layers, with the last layer consisting of a single neuron with bias $b_L \in \mathbb{R}$, and weight vector $\boldsymbol{w}_L$. For any other layer $l$, its bias vector is $\boldsymbol{b}_l$ and weight matrix is $\boldsymbol{W}_l$. We use the notation $[n]$ to denote the set $\{1, \ldots, n\}$.

We formally define memorization as follows:

**Definition 1.** *A learning algorithm $\mathcal{A}$ can memorize a dataset of feature vectors $\mathcal{D} = \{\boldsymbol{x}_i\}_{i=1}^n$ if for any arbitrary labeling $\{y_i\}_{i=1}^n$ of the dataset, the algorithm can output a model $f$ such that $\forall i \in [n] : f(\boldsymbol{x}_i) = y_i$.*

For instance, $\mathcal{A}$ could be the training procedure for a particular neural architecture and $f$ could be the neural network that $\mathcal{A}$ outputs after the training process. For neural networks, we will abuse the notation and say that a neural network $f$ can memorize a dataset $\mathcal{D} = \{\boldsymbol{x}_i\}_{i=1}^n$ if for every arbitrary labeling $\{y_i\}_{i=1}^n$ of the dataset, we can find a set of weights and biases for the network so that $\forall i \in [n] : f(\boldsymbol{x}_i) = y_i$.

We are mainly interested in the minimum size of a threshold network so that it can memorize a dataset of $n$ samples. Our only assumption on the dataset will be that of $\delta$-separation, which we define below.

**Definition 2.** *We say that a dataset is $\delta$-separated if it satisfies either Assumption 1 **or** Assumption 2.*

**Assumption 1.** *(Angular separation) All the feature vectors in the dataset, $\{\boldsymbol{x}_i\}_{i=1}^n$ satisfy*

$$\arccos\left(\frac{\langle \boldsymbol{x}_i, \boldsymbol{x}_j \rangle}{\|\boldsymbol{x}_i\|\|\boldsymbol{x}_j\|}\right) \geq \delta$$

**Assumption 2.** *(Normalized minimum distance) All the feature vectors in the dataset, $\{\boldsymbol{x}_i\}_{i=1}^n$ satisfy $\|\boldsymbol{x}_i\| \leq 1$ and*

$$\forall i \neq j : \|\boldsymbol{x}_i - \boldsymbol{x}_j\| \geq \delta.$$

**Remark 1.** *Note that the 'normalized' in the name of Assumption 2 refers to the fact that the maximum norm of any point in the dataset is bounded by 1. Our results can be easily extended to any other bounded dataset by rescaling the weights and biases of the first layer appropriately. Park et al. (2020) also use a similar normalized distance assumption.*

Note that if the points lie on the unit sphere, like in Vershynin (2020) or Theorem 2 in this paper, then both assumptions are roughly equivalent.

**Notation:** We use the lower case letters for scalars ($w$), lower case bold letters for vectors ($\boldsymbol{w}$), and upper case bold letters for matrices ($\boldsymbol{W}$). We denote the $i$-th element of vector $\boldsymbol{w}$ as $w_i$, similarly for the case when the vectors have subscript, we denote the $i$-th element of $\boldsymbol{w}_j$ by $w_{j,i}$. We use a tilde over $\mathcal{O}(\cdot)$, $\Omega(\cdot)$ and $\Theta(\cdot)$, *i.e.*, $\widetilde{\mathcal{O}}(\cdot)$, $\widetilde{\Omega}(\cdot)$ and $\widetilde{\Theta}(\cdot)$, to hide the logarithmic factors.

## 4 Main results

As discussed earlier, the existing result by Vershynin (2020) has an exponential dependence on $\delta$. In this section, we provide a new and tighter upper bound on the number of weights and neurons needed for a memorization, which brings the dependence down to almost linear. Our result is stated below:

**Theorem 1.** *There exists a threshold activated neural network with $\mathcal{O}(\frac{\log^2 n}{\delta} + \sqrt{n}\log^2 n)$ neurons and $\mathcal{O}\left(\frac{(d+\log n)\log n}{\delta} + n\log^2 n\right)$ weights that can memorize any dataset of size $n$ in $d$ dimensions that is $\delta$-separated.*

In this construction, only the first layer has real weights. The rest of the network has binary and integer weights. In particular, the $O(n\log^2 n)$ term in the number of weights comes from binary and integer weights (with the integers being bounded and on the order of $\mathcal{O}(\log n)$). Hence, the $\mathcal{O}(n\log^2 n)$

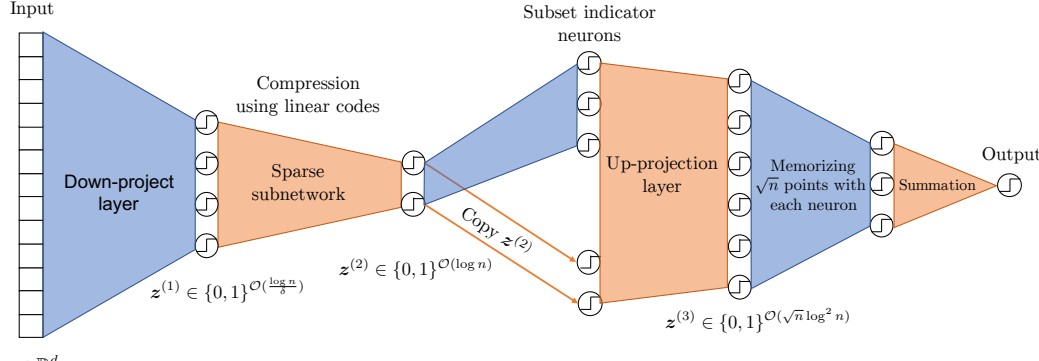

Figure 1: The schematic of the neural network that can memorize any $\delta$-separated dataset of size $n$ in $d$ dimensions (Theorem 1). The first layer has random Gaussian weights and projects the input from $\mathbb{R}^d$ down to $\{0,1\}^{\mathcal{O}(\frac{\log n}{\delta})}$. The next component is a sparse subnetwork that further compresses the vectors from $\{0,1\}^{\mathcal{O}(\frac{\log n}{\delta})}$ to $\{0,1\}^{\mathcal{O}(\log n)}$ using linear codes. The next couple of layers lift the vectors from $\{0,1\}^{\mathcal{O}(\log n)}$ to $\{0,1\}^{\mathcal{O}(\sqrt{n}\log^2 n)}$ while ensuring that we can find sets of vectors of size $\sqrt{n}$ that are linearly independent. Finally, we use this linear independence to memorize up to $\sqrt{n}\log^2 n$ points with a single neuron. This gives us a layer of size $\sqrt{n}$ that can memorize the entire dataset. The final layer is just a single neuron that sums the outputs of these $\sqrt{n}$ neurons.

weights can actually be represented by $\mathcal{O}(n\log^2 n \log\log n)$ bits. We will see in Section 6 that $\Omega(n)$ bits are necessary for memorization by any model and hence this term is optimal, up to logarithmic factors. The $\frac{d\log n}{\delta}$ weights in the bound are the weights of the first layer which has width $\frac{\log n}{\delta}$, for which later, in Theorem 2, we will provide some indication that this factor might also be necessary.

Baum (1988) had constructed a dataset with normalized minimum distance $\mathcal{O}(1/n)$ and proved that for this dataset, any threshold network aiming to memorize this must have $\lceil n/d \rceil$ neurons. We introduce a new, $\delta$-dependent lower bound below:

**Theorem 2.** *For every $0 < \delta \leq \frac{1}{2}$ and some universal constants $C_1$ and $C_2$, there exists a dataset of size $n \in \left[ \frac{C_1 d^2 \log^2(d/\delta)}{\delta}, \left(\frac{C_2}{\delta}\right)^{\frac{d}{2}} \right]$ satisfying the $\delta$-separation condition such that any threshold network that can memorize it needs $\Omega\left(\frac{1}{\sqrt{\delta}\log(1/\delta)}\right)$ neurons in the first layer.*

Note that the dataset in this theorem actually satisfies *both* Assumption 1 and Assumption 2, up to a constant factor. To understand the implication of Theorem 2 a bit better, consider two dataset $\mathcal{D}_1$ and $\mathcal{D}_2$, both containing $n$ points on the sphere. We assume that $\mathcal{D}_1$ has $\delta_1$-separation and $\mathcal{D}_2$ has $\delta_2$-separation, where $\delta_1 = \Theta(n^{-2/d})$ and $\delta_2 = \widetilde{\Theta}(d^2/n)$. Then, Theorem 1 says that there exists a network that can memorize $\mathcal{D}_1$ with only $\mathcal{O}(n^{2/d}\log n)$ neurons in the first layer, whereas Theorem 2 says that any network aiming to memorize $\mathcal{D}_2$ would need at least $\widetilde{\Omega}(\sqrt{n}/d)$ neurons in the first layer. Hence, we see that the minimum distance can have a big impact on the network architecture.

Comparing Theorems 1 and 2, we see that there is a gap in the bounds: Theorem 1 proves an upper bound of $\widetilde{\mathcal{O}}(1/\delta)$ neurons, whereas Theorem 2 proves a lower bound of $\widetilde{\Omega}(1/\sqrt{\delta})$ neurons. We think that the gap is an artifact of the proof of Theorem 2, and a tighter analysis or a better construction could result in a lower bound of $\widetilde{\Omega}(1/\delta)$ neurons, which will match the upper bound of Theorem 1. However, this is just speculation based on intuition, and more research will be needed to answer this open question.

## 5 Memorization with threshold networks

In this section, we give a proof sketch of Theorem 1, the complete proof is provided in the appendix. We will construct a network that can memorize any $\delta$-separated dataset. As we discussed before,

the threshold activation prohibits the passing of amplitude information to deeper layers. This is the biggest obstacle in leveraging the benefits of multiple layers of transformation that can be obtained by networks with activations such as ReLU. In the following, we will explain our construction layer-wise:

**Step 1: Generating unique binary representations.** Given that the amplitude information is lost after thresholding, we want to ensure that at the least, the first layer is able to transform the inputs into binary representations such that each sample in our dataset has a unique binary representation. Later, we will see that this is sufficient for memorization.

Now we give an overview of how a layer with $\mathcal{O}(\frac{\log n}{\delta})$ neurons with random i.i.d. weights can convert the input into unique binary vectors of length $\mathcal{O}(\frac{\log n}{\delta})$. In this sketch, we will do this under Assumption 1, but the same technique can be extended for Assumption 2.

The key result we use is that any hyperplane passing through the origin, with random Gaussian coefficients has at least a $\delta/2\pi$ probability of separating two points which have an angle of at least $\delta$ between them. To see how this is true, consider two points $\boldsymbol{x}_i$ and $\boldsymbol{x}_j$ with an angle $\delta$ between them. Consider the 2-dimensional space spanned by these two vectors. Note that the intersection of the Gaussian hyperplane with the 2-dimensional space spanned by $\boldsymbol{x}_i$ and $\boldsymbol{x}_j$ is just a line passing through the origin. Further, because the Gaussian hyperplane has isometric coefficients, it can be shown that the line has angle uniformly distributed in $[0, 2\pi)$. This implies that the probability that this line passes in between the two points is at least $\delta/2\pi$.

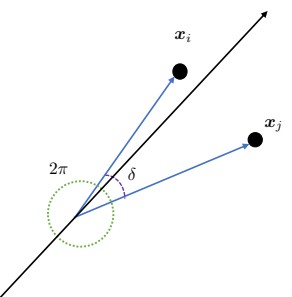

If we take $m$ such independent Gaussian hyperplanes, then the probability that none of them separate a pair $\boldsymbol{x}_i, \boldsymbol{x}_j$ is less than $(1 - \frac{\delta}{2\pi})^m$. Finally, taking a simple union bound over the $\binom{n}{2}$ pair of points, shows that the probability that at least one pair has no hyperplane passing in between them is at most

$$\binom{n}{2}\left(1 - \frac{\delta}{2\pi}\right)^m.$$

We want this to be less than $1$ to show that there exists one set of hyperplanes which can separate all the pairs. Doing that gives us $m = \mathcal{O}(\frac{\log n}{\delta})$, which is our required bound.

For any input $\boldsymbol{x}_i$, each threshold neuron in the first layer outputs either 0 or 1. We can concatenate the outputs of all the neurons in the first layer into a binary vector. Let us denote this binary representation of $\boldsymbol{x}_i$ created by the first layer as $\boldsymbol{z}_i^{(1)}$, for $i = 1, \ldots, n$. Further, denote the dimension of $\boldsymbol{z}_i^{(1)}$ by $d^{(1)}$, which is equal to the number neurons in the first layer, that is, $d^{(1)} = \mathcal{O}(\frac{\log n}{\delta})$.

Figure 2: The probability that a uniformly random line passing through the origin passes in between $\boldsymbol{x}_i$ and $\boldsymbol{x}_j$ is $\delta/2\pi$.

**Step 2: Compressing the binary representations to optimal length.** In order to avoid a factor $\delta$ in the later layers, our next task is to further compress the $\mathcal{O}(\frac{\log n}{\delta})$ dimensional vectors returned by the first layer into vectors of length $\mathcal{O}(\log n)$. Note that $\mathcal{O}(\log n)$ length is optimal (up to constant multiplicative factors) if we want to represent the $n$ samples uniquely using binary vectors. For this we will use linear codes where we simulate XOR operations by threshold activations.

Consider two $d^{(1)}$-dimensional binary vectors $\boldsymbol{z}_i^{(1)}$ and $\boldsymbol{z}_j^{(1)}$ that differ in at least one bit. Let $\oplus$ represent the XOR operation. Let $\boldsymbol{b}$ be a random binary vector of length $d^{(1)}$ where each bit is i.i.d Bernoulli(0.5). Then, we claim that

$$\Pr(\langle \boldsymbol{z}_i^{(1)}, \boldsymbol{b}\rangle_\oplus \neq \langle \boldsymbol{z}_j^{(1)}, \boldsymbol{b}\rangle_\oplus) = 0.5,$$
$$\text{where } \langle \boldsymbol{z}, \boldsymbol{b}\rangle_\oplus = (z_1 \cdot b_1) \oplus \cdots \oplus (z_{d^{(1)}} \cdot b_{d^{(1)}})$$

For the ease of exposition, assume that $\boldsymbol{z}_i^{(1)}$ and $\boldsymbol{z}_j^{(1)}$ differ in only the first bit (the proof of the general case is provided in the appendix). Let the first bit of $\boldsymbol{z}_i^{(1)}$ be 0, and that of $\boldsymbol{z}_j^{(1)}$ be 1, and we have assumed that the rest of their bits are the same. Let $c := (z_{i,2}^{(1)} \cdot b_2) \oplus \cdots \oplus (z_{i,d^{(1)}}^{(1)} \cdot b_{d^{(1)}}) =$

$(z_{j,2} \cdot b_2) \oplus \cdots \oplus (z_{j,d^{(1)}} \cdot b_{d^{(1)}})$. Then,

$$\Pr(\langle \boldsymbol{z}_i^{(1)}, \boldsymbol{b} \rangle_\oplus \neq \langle \boldsymbol{z}_j^{(1)}, \boldsymbol{b} \rangle_\oplus) = \Pr(((0 \cdot b_1) \oplus c) \neq ((1 \cdot b_1) \oplus c)).$$

It can now be verified that if $b_1 = 1$, then $((0 \cdot b_1) \oplus c) \neq ((1 \cdot b_1) \oplus c)$. Since $\Pr(b_1 = 1) = 0.5$, we get the desired result.

We have shown that the probability that one such random vector $\boldsymbol{b}$ differentiates between a pair $\boldsymbol{z}_i^{(1)}$ and $\boldsymbol{z}_j^{(1)}$ with probability 0.5. Doing an analysis similar to the one that we did for the first layer, we get that $m = \mathcal{O}(\log n)$ vectors suffice to separate all pairs. The big task here, however, is to implement the XOR operation using threshold activations. In the appendix, we show that this can in fact be achieved using a couple of sparse layers.

Similar to the output of the first layer, let us denote the compressed binary representation of $\boldsymbol{z}_i^{(1)}$ created by these sparse layers as $\boldsymbol{z}_i^{(2)}$, for $i = 1, \ldots, n$. Further, let $\boldsymbol{z}_i^{(2)}$ have dimension $d^{(2)}$, where $d^{(2)} = \mathcal{O}(\log n)$, as we showed above.

**Step 3: Partitioning the samples into subsets of size $\mathcal{O}(\sqrt{n} \log n)$.** Once we have $d^{(2)} = \mathcal{O}(\log n)$ length binary representations of the inputs, memorizing with a single layer consisting of $(d^{(2)} + 1)n$ weights and $n$ neurons is not difficult. However, our aim is to reduce the number of neurons to $\mathcal{O}(\sqrt{n} \log n)$. For that, we use a strategy similar to Yun et al. (2019): we will memorize up to $K$ samples with 1 neuron, where $K = \Omega(\sqrt{n} \log n)$. Thus, we will need only $\mathcal{O}(\sqrt{n})$ neurons in total. Roughly speaking the strategy would be the following: Let's say we have a neuron $\sigma(\boldsymbol{w}^\top \boldsymbol{z} + b)$. Then, if for any $\{y_1, \ldots, y_K\} \in \{0,1\}^K$, we can find a $\boldsymbol{w}$ and $b$ such that all of the following equations hold at the same time,

$$\boldsymbol{w}^\top \boldsymbol{z}_1 + b = y_1, \quad \ldots \quad, \boldsymbol{w}^\top \boldsymbol{z}_K + b = y_K, \quad (1)$$

then this neuron can memorize the $K$ samples $\{\boldsymbol{z}_1, \ldots \boldsymbol{z}_K\}$. To do so, we need that the vectors $\{\boldsymbol{z}_1, \ldots, \boldsymbol{z}_K\}$ be linearly independent[4]. Since we have $K = \Omega(\sqrt{n} \log n)$, we require that $\boldsymbol{z}_i$'s have dimension at least $K$. However, the previous few layers have compressed the samples into $d^{(2)} = \mathcal{O}(\log n)$ length binary vectors. Thus, the next task is to project these $d^{(2)}$ length binary vectors to $\Omega(K)$ length binary vectors. Note that first compressing down to $\mathcal{O}(\log n)$ length vectors and then expanding these up to $\Omega(\sqrt{n} \log n)$ length binary vectors seems counter intuitive. However, we were unable to project the vectors directly to $\Omega(\sqrt{n} \log n)$ length binary vectors without incurring a large dependence on $\delta$ and $n$ in the number of weights and neurons.

One more objective that we accomplish while projecting up to $\Omega(K)$ dimension is that both the projection and memorizing the resulting vectors can be easily done with bounded integer weights. The way we do this is by partitioning the $n$ vectors $\boldsymbol{z}_i^{(2)}$ into $K$ subsets of size at most $\sqrt{n}$ each. We prove that we can find such $K$ subsets, such

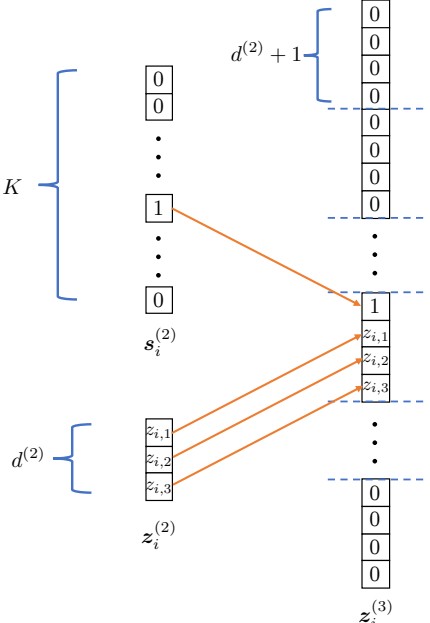

Figure 3: In **Step 3**, $\boldsymbol{z}_i^{(2)}$ is copied to a location in $\boldsymbol{z}_i^{(3)}$ depending on the subset $S_j$ to which $\boldsymbol{z}_i^{(2)}$ belongs.

that each is characterized by a unique prefix, which all the elements of that subset share. These prefixes ensure that we can easily detect which subset a particular $\boldsymbol{z}_i^{(2)}$ belongs to, using threshold neurons. For more details on this, please refer to the appendix.

---

[4]Yun et al. (2019) also use this strategy. However, as we will see in Step 4, we are able to ensure that the vectors $\{\boldsymbol{z}_1, \ldots, \boldsymbol{z}_K\}$ are linearly independent while also ensuring that their coordinates do not overlap. This helps us find a vector $\boldsymbol{w}$ which has integer values and satisfies (1). Further, since Yun et al. (2019) use ReLU activations, the way they achieve linear independence, and the construction of the rest of the network is different.

Let $\{S_1, \ldots, S_K\}$ denote the $K$ subsets, each with a unique prefix. Let $\boldsymbol{z}_i^{(2)} \in \{0,1\}^{d^{(2)}}$ be the binary vector returned by **Step 2** above for input $\boldsymbol{x}_i$. We mentioned above that by using threshold neurons, we can create a vector $\boldsymbol{s}_i^{(2)} \in \{0,1\}^K$ for $\boldsymbol{z}_i^{(2)}$ such that $s_{i,j} = 1$ if and only if $\boldsymbol{z}_i^{(2)} \in S_j$. Then, using these prefixes and threshold neurons, we can create the expanded binary vector $\boldsymbol{z}_i^{(3)} \in \{0,1\}^{K(d^{(2)}+1)}$ corresponding to the vector $\boldsymbol{z}_i^{(2)}$. The way we create $\boldsymbol{z}_i^{(3)}$ from $\boldsymbol{z}_i^{(2)}$ is as follows: $\boldsymbol{z}_i^{(3)}$ is partitioned into $K$ chunks of length $d^{(2)} + 1$. If $\boldsymbol{z}_i^{(2)} \in S_j$, then the first bit of the $j$-th chunk is set to 1 and $\boldsymbol{z}_i^{(2)}$ is copied to the rest of the bits in that chunk. All the other bits in $\boldsymbol{z}_i^{(3)}$ are 0. This is shown in Figure 3. This operation can be done with threshold activated neuron, for details please refer to the appendix.

**Step 4: Memorizing the samples with $\sqrt{n}$ neurons** A key property about the vectors $\boldsymbol{z}_i^{(3)}$ is that if we take $K$ such vectors, each belonging to a different subset $S_j$, then they will be linearly independent as their coordinates do not overlap. Thus, we can find $\boldsymbol{w}$ and $b$ such that (1) is satisfied. Furthermore, due to the fact that these $K$ vectors have different supports, we can manage to satisfy (1) with $\boldsymbol{w}$ which have integer elements on the order of $\mathcal{O}(\log n)$.

Hence, we can memorize up to $K$ samples with one neuron. This gives that we need $\sqrt{n}$ in the second-to-last layer to memorize all the samples. The final layer is simply a single neuron that sums the outputs of the second-to-last layer.

Note that a threshold neuron can be approximated using three ReLU neurons. Hence, the construction described in this section can also be used to create a ReLU activated network with can memorize the same dataset.

## 5.1 Extending the construction to multiple class memorization

The construction above can be easily extended to the case when there are more than two classes. Let there be $M \geq 2$ classes. Then, the label for each sample can be represented as a sequence of $\log_2 M$ bits. In this case, we can replicate the construction of the last and the second-to-last layers $\log_2 M$ times so that the new network has $\log_2 M$ output neurons, each output neuron memorizing one bit of the label. Hence, we get the following Corollary:

**Corollary 1.** *There exists a threshold activated neural network with $\mathcal{O}(\frac{\log^2 n}{\delta} + \sqrt{n}(\log^2 n)(\log M))$ neurons and $\mathcal{O}\left(\frac{(d+\log n)\log n}{\delta} + n(\log^2 n)(\log M)\right)$ weights that can memorize any $M$-class dataset of size $n$ in $d$ dimensions that is $\delta$-separated.*

# 6 Bit complexity of memorization

Before understanding the minimum size that threshold networks have to be in order to memorize a dataset, a more fundamental question is how many bits are needed to specify the models output by any learning algorithm that aims to memorize a dataset. In this section, we provide two theorems that sharply characterize the bit complexity needed for memorization. The results provide the bounds in the terms of two very related quantities, the *packing number* and the *covering number*, which we define below.

**Definition 3.** *Given a real number $\delta > 0$, the packing number $\mathcal{P}_\delta$ of a set $\mathcal{S}$ (equipped with norm $\|\cdot\|$) is defined as the cardinality of the largest set $\mathcal{X} \subset \mathcal{S}$ such that*

$$\forall x_1, x_2 \in \mathcal{X} : \|x_1 - x_2\| \geq \delta.$$

Intuitively, the packing number is the maximum number of $\delta/2$ radius non-overlapping balls that one can fit inside a set.

**Definition 4.** *Given a real number $\delta > 0$, the covering number $\mathcal{C}_\delta$ of a set $\mathcal{S}$ (equipped with norm $\|\cdot\|$) is defined as the cardinality of the smallest set $\mathcal{X} \subset \mathcal{S}$ such that*

$$\forall x \in \mathcal{S}, \exists \hat{x} \in \mathcal{X} : \|x - \hat{x}\| \leq \delta.$$

Intuitively, the covering number is the minimum number of $\delta$ radius (possibly overlapping) balls that one needs to completely cover a set.

These quantities are very useful in quantifying the 'size' of a set. These are related to each other through the following relation (Vershynin, 2018, Lemma 4.2.8):

$$\mathcal{P}_{2\delta} \leq \mathcal{C}_\delta \leq \mathcal{P}_\delta.$$

Now, we are ready to state the upper and lower bounds on the bit complexity of memorization.

**Theorem 3.** *Let $\mathcal{S}$ be a set from which we select the samples. Let $\mathcal{A}$ be a learning algorithm that can memorize any dataset of feature vectors $\mathcal{D} \in \mathcal{S}^n$ as long as every pair of feature vectors $\boldsymbol{x}_i, \boldsymbol{x}_j \in \mathcal{D}$ satisfies $\|\boldsymbol{x}_i - \boldsymbol{x}_j\| \geq \delta$. Then, we need at least $\max(n, \log_2 \log_2 \mathcal{P}_\delta)$ bits to represent the models output by $\mathcal{A}$.*

**Theorem 4.** *Let $\mathcal{S}$ be a set from which we select the samples. Then, there exists a learning algorithm $\mathcal{A}$ that can memorize any dataset of feature vectors $\mathcal{D} \in \mathcal{S}^n$ as long as every pair of feature vectors $\boldsymbol{x}_i, \boldsymbol{x}_j \in \mathcal{D}$ satisfies $\|\boldsymbol{x}_i - \boldsymbol{x}_j\| \geq \delta$, and the models returned by $\mathcal{A}$ can be represented in $O(n + \log_2 \log_2 \mathcal{C}_{\delta/2})$ bits.*

Note that since $a + b = \mathcal{O}(\max(a, b))$ for non-negative variables $a$ and $b$, the upper bound from Theorem 4 is of the same order as the lower bound from Theorem 3.

It is known that the packing and covering numbers of the unit sphere are of the order $\Theta((C_{\mathsf{p}}/\delta)^d)$ and $\Theta((C_{\mathsf{c}}/\delta)^d)$, for some universal constants $C_{\mathsf{p}}$ and $C_{\mathsf{c}}$ respectively. Substituting this into Theorems 3 and 4, we get the following corollaries.

**Corollary 2.** *Let $\mathcal{S}^{d-1}$ be the unit sphere in $\mathbb{R}^d$. Let $\mathcal{A}$ be a learning algorithm that can memorize any set of $n$ points on $\mathcal{S}^{d-1}$ as long as every pair of points $\boldsymbol{x}_i, \boldsymbol{x}_j \in \mathcal{D}$ satisfies $\|\boldsymbol{x}_i - \boldsymbol{x}_j\| \geq \delta$. Then, we need $\Omega(\max(n, \log d + \log \log \frac{1}{\delta}))$ bits to represent the models output by $\mathcal{A}$.*

**Corollary 3.** *Let $\mathcal{S}^{d-1}$ be the unit sphere in $\mathbb{R}^d$. Then, there exists a learning algorithm $\mathcal{A}$ that can memorize any set of $n$ points on $\mathcal{S}^{d-1}$ as long as every pair of feature vectors $\boldsymbol{x}_i, \boldsymbol{x}_j \in \mathcal{D}$ satisfies $\|\boldsymbol{x}_i - \boldsymbol{x}_j\| \geq \delta$, and the models returned by $\mathcal{A}$ can be represented in $O(n + \log d + \log \log \frac{1}{\delta})$ bits.*

Note that if we have a set of $n$ points on the unit sphere separated by at least distance $\delta$, then the set satisfies *both* Assumption 1 and Assumption 2. Hence, these two corollaries give bounds on the bit complexity of memorization under both the $\delta$-separation assumptions.

## 7 Lower bound construction

For our lower bound construction, we will have $n$ points on the unit sphere in $d$-dimensions. Each neuron in the first layer represents a hyperplane in $d$-dimensions. The main observation that we use is that for every pair of points $\boldsymbol{x}_i$ and $\boldsymbol{x}_j$ such that they have different labels, there should be a hyperplane passing in between them. Thus, we aim to lower bound the minimum number of hyperplanes needed for this to happen. This is related to the problem of having to separate every point using hyperplanes, that is, the problem where we need to ensure that there is a hyperplane passing in between *every* pair of points $\boldsymbol{x}_i$ and $\boldsymbol{x}_j$. For this problem we have the following theorem.

**Theorem 5.** *For every $0 < \delta \leq \frac{1}{2}$ and some universal constants $C_1$ and $C_2$, there exists a set of $n$ points on the $d$-dimensional sphere, with $n \in \left[ \frac{C_1 d^2 \log^5(d/\delta)}{\delta}, \left(\frac{C_2}{\delta}\right)^{\frac{d}{2}} \right]$, such that each pair of points is separated by a distance of at least $\delta$, and one needs $\Omega\left(\frac{\log n}{\sqrt{\delta} \log \frac{\log n}{\delta}}\right)$ hyperplanes to separate them.*

We use essentially the same construction for both the problems. On the sphere, first we sample $\sqrt{n}$ points uniformly at random, which we call *cluster centers*. Around

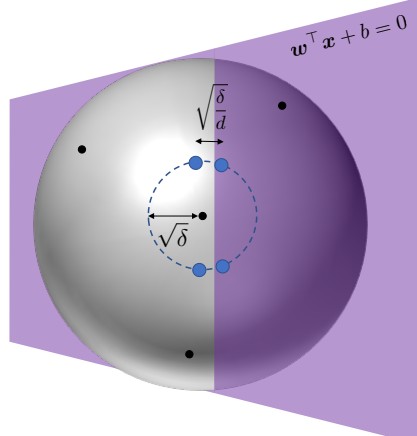

Figure 4: In this figure, the black dots represent the cluster centers and the blue circles represent the samples in a cluster. In high dimensions, the points in each cluster get concentrated within distance $\mathcal{O}(\sqrt{\delta/d})$ in any direction. Hence any hyperplane, that wants to separate a constant fraction of the points in a cluster, needs to pass within distance $\mathcal{O}(\sqrt{\delta/d})$ of the cluster center, as shown in the figure.

each of these centers, we sample $\sqrt{n}$ points uniformly at random from the cap of radius $\sqrt{\delta}$, and call these sets of points *clusters*. Hence, in total we sample $n$ points on the sphere and it can be shown that with high probability these will all be at least $\delta$ distance away from each other as long as $n \in \left[\frac{d^2}{\delta}, \frac{1}{\delta^{d/2}}\right]$. In high dimensions, it can be shown that for each cluster and in any direction, most points lie within a distance $\sqrt{\delta/d}$ of the cluster center. Hence, if a hyperplane aims to 'effectively' separate pairs of points in a cluster, it needs to pass within distance $\sqrt{\delta/d}$ of the cluster center. It can also be shown that in high dimensions any hyperplane can only pass within distance $\sqrt{\delta/d}$ of the cluster centers of $\sqrt{\delta}$ fraction of the clusters. Finally, we use the following necessary inequality to get a bound on the required number of hyperplanes:

$$(\# \text{ hyperplanes needed})$$
$$\geq \frac{(\# \text{ clusters}) \times (\# \text{ hyperplanes needed per cluster})}{(\# \text{ clusters a hyperplane can 'effectively' separate})}.$$

We know that the number of clusters is $\sqrt{n}$, and we showed above that ($\#$ clusters a hyperplane can 'effectively' separate) is $\sqrt{\delta}$ fraction of the clusters, that is $\mathcal{O}(\sqrt{n\delta})$. Finally, for the term ($\#$ hyperplanes needed per cluster), we use 1 for the memorization problem, where we need to separate points with opposite label. For the problem of separating every pair of points, we know that ($\#$ hyperplanes needed per cluster) $= \Omega(\log_2 \sqrt{n})$ since there $\sqrt{n}$ points in each cluster and hence we need at least $\log_2 \sqrt{n}$ hyperplanes per cluster. Substituting these values gives us the bounds in Theorems 2 and 5.

Note that we skipped many details in the sketch above. The complete proof is provided in the appendix, and the construction there is slightly different.

# 8 Conclusion

In this work, we study the memorization capacity of threshold networks under the $\delta$-separation assumption. We improve the existing bounds on the number of neurons and weights required, from exponential in $\delta$, to almost linear. We also prove new, $\delta$-dependent lower bounds for the memorization capacity of threshold networks, that together with our upper bound, shows that $\delta$ (the separation) indeed impacts the network architecture required for memorization.

## Acknowledgments and Disclosure of Funding

DP acknowledges the support of NSF CAREER Award #1844951, Sony Faculty Innovation Award, AFOSR & AFRL Center of Excellence Award FA9550-18-1-0166, NSF TRIPODS Award #1740707, and ONR YIA Grant N00014-21-1-2806. AK acknowledges the support of NSF CAREER Award IIS-1845032, and ONR Grant N00014-19-1-240.

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
