## A    Comparing Vershynin (2020) and Baum (1988)

In this section, we show how the assumptions in Vershynin (2020) and Baum (1988) are not comparable, and in different situations one result can be better than the other.

First we show an example where the points are well-separated, but do not lie in general position. Let $d = \Omega(\log n)$. Note that $n$ points can be arranged in a $\mathcal{O}(\log n)$-dimensional unit sphere while still ensuring that the distance between them is a constant, that is, $\delta$ is a universal constant. Hence, even though the ambient space is $d$-dimensional, the dataset can lie on a $\mathcal{O}(\log n)$-dimensional sphere, and be separated by a constant distance. To visualize this, consider how in a 3-dimensional space, a dataset could lie on a circle (which is 2-dimensional). In this setting, Vershynin's bound would still need $\widetilde{\mathcal{O}}(\sqrt{n})$ neurons and $\widetilde{\mathcal{O}}(n + d)$ weights. On the other hand, naively extending Baum's construction to this setting would require $\mathcal{O}(n/\log n)$ neurons and thus $\mathcal{O}(nd/\log n)$ weights.

Next, consider the case when the points lie in general position, but the distance between them is infinitesimally small. In this case, Vershynin's bounds containing the $e^{1/\delta^2}$ term will be much larger than Baum's bounds which are independent of minimum distance.

Finally, if the points are indeed well separated *and* in general position, then Vershynin's construction will be better when $d \ll \sqrt{n}$, and Baum's construction will be better when $d \gg \sqrt{n}$.

Note that the comparison between our results and Baum's results would be identical to the comparison done above.

## B    Proof of Theorem 1

*Proof.*  As explained in Section 5, the network memorizes in four steps. In this proof, we will go into the details of each step.

**Step 1: Generating unique binary representations.**    The first layer's task is to generate unique binary representations for each sample in the dataset. What this means is that for every pair of samples $\boldsymbol{x}_i$ and $\boldsymbol{x}_j$, there should exist at least one neuron in the first layer, say $\sigma(\boldsymbol{w}^\top \boldsymbol{x} + b)$ such that $\sigma(\boldsymbol{w}^\top \boldsymbol{x}_i + b) \neq \sigma(\boldsymbol{w}^\top \boldsymbol{x}_j + b)$. Note that $\boldsymbol{w}^\top \boldsymbol{x} + b$ is just a hyperplane and $\sigma(\boldsymbol{w}^\top \boldsymbol{x}_i + b) \neq \sigma(\boldsymbol{w}^\top \boldsymbol{x}_j + b)$ is equivalent to saying that $\boldsymbol{x}_i$ and $\boldsymbol{x}_j$ lie on the opposite sides of the hyperplane. The next two lemmas says that we can easily find a set of $\mathcal{O}(\log(n)/\delta)$ hyperplanes such that for every pair of points $\boldsymbol{x}_i, \boldsymbol{x}_j$, at least one hyperplanes passes in between them.

**Lemma 1.**  *Assume that our dataset satisfies Assumption 1. Define $m := \left\lceil \frac{4\pi}{\delta} \log\left(\frac{n}{\epsilon}\right) \right\rceil$ for any $\epsilon \in (0, 1)$. If we sample $m$ hyperplanes of the form $\mathbf{w}_i^\top \boldsymbol{x} = 0$, for $i = 1, \ldots, m$, where $\mathbf{w}_i \overset{i.i.d.}{\sim} \mathcal{N}(\mathbf{0}, \boldsymbol{I}_d)$, then with probability $1 - \epsilon$,*

$$\forall i \neq j, \exists k : (\mathbf{w}_k^\top \boldsymbol{x}_i)(\mathbf{w}_k^\top \boldsymbol{x}_j) < 0.$$

**Lemma 2.**  *Assume that our dataset satisfies Assumption 2. Define $m := \frac{C}{\delta} \log\left(\frac{n}{\epsilon}\right)$ for some universal constant $C$ and any $\epsilon \in (0, 1)$. If we sample $m$ hyperplanes of the form $\mathbf{w}_i^\top \boldsymbol{x} + \mathbf{b}_i = 0$, for $i = 1, \ldots, m$, where $\mathbf{w}_i \overset{i.i.d.}{\sim} \mathcal{N}(\mathbf{0}, \boldsymbol{I}_d)$ and $\mathbf{b}_i \overset{i.i.d.}{\sim} \mathcal{N}(0, 1)$, then with probability $1 - \epsilon$,*

$$\forall i \neq j, \exists k : (\mathbf{w}_k^\top \boldsymbol{x}_i + \mathbf{b}_k)(\mathbf{w}_k^\top \boldsymbol{x}_j + \mathbf{b}_k) < 0.$$

Note that each neuron in the first layer outputs either 0 or 1. If we concatenate all the outputs of these $\mathcal{O}(\log(n)/\delta)$ neurons, we will get a binary vector of length $\mathcal{O}(\log(n)/\delta)$. The lemmas above say that under Assumption 1 *or* Assumption 2, these binary vectors will be unique.

Let $\boldsymbol{z}_i^{(1)}$ be the binary vector created by the first layer when the input to the network is $\boldsymbol{x}_i$. Further, let $d^{(1)}$ denote the dimension (length) of this binary vector. Note that the first layer has $\mathcal{O}(\log(n)/\delta)$ neurons and $\mathcal{O}(d\log(n)/\delta)$ weights and biases.

**Step 2: Compressing the binary representations.**    The next few layers compress the $\mathcal{O}(\log(n)/\delta)$ length binary vectors down to $\mathcal{O}(\log(n))$ length unique binary representations for each sample. We will use linear codes in $\mathbb{F}_2$ (the finite field of two elements: 0 and 1) for this.

For two binary vectors $\boldsymbol{b}_1, \boldsymbol{b}_2 \in \{0,1\}^{d^{(1)}}$, define the inner product in the field as

$$\langle \boldsymbol{b}_1, \boldsymbol{b}_2 \rangle_\oplus := (b_{1,1} \cdot b_{2,1}) \oplus (b_{1,2} \cdot b_{2,2}) \oplus \cdots \oplus (b_{1,d^{(1)}} \cdot b_{2,d^{(1)}}),$$

where $\oplus$ is the XOR operator, and $b_{1,i}$ is the $i$-th element of $\boldsymbol{b}_1$, and similarly $b_{2,i}$ is the $i$-th element of $\boldsymbol{b}_2$.

Let $\mathbf{b}_1, \ldots, \mathbf{b}_m$ be $m$ i.i.d. Bernoulli(0.5) vectors of length $d^{(1)}$ each. Define vectors $\boldsymbol{z}_i^{(2)}$ as

$$\boldsymbol{z}_i^{(2)} = \begin{bmatrix} \langle \mathbf{b}_1, \boldsymbol{z}_i^{(1)} \rangle_\oplus \\ \vdots \\ \langle \mathbf{b}_m, \boldsymbol{z}_i^{(1)} \rangle_\oplus \end{bmatrix}, \forall i \in [n]. \tag{2}$$

The next lemma says that we only need $m = \mathcal{O}(\log n)$ to ensure that $\forall i \neq j, \boldsymbol{z}_i^{(2)} \neq \boldsymbol{z}_j^{(2)}$.

**Lemma 3.** *Let $\boldsymbol{z}_1^{(1)}, \ldots, \boldsymbol{z}_n^{(1)}$ be $n$ distinct binary vectors of dimension $d^{(1)}$, and let $\mathbf{b}_1, \ldots, \mathbf{b}_m$ be $m$ vectors in $d^{(1)}$ dimensions, with i.i.d. Bernoulli(0.5) entries. Let $\boldsymbol{z}_1^{(2)}, \ldots, \boldsymbol{z}_n^{(2)}$ be as defined in (2). If $m = 3 \log \frac{n}{\epsilon'}$, then with probability $1 - \epsilon'$,*

$$\forall i \neq j : \boldsymbol{z}_i^{(2)} \neq \boldsymbol{z}_j^{(2)}.$$

The lemma above is a special case of the Gilbert-Varshamov bound (for example, see (Guruswami et al., 2012)). A simple proof of the lemma is provided in subsection B.1.3 for completeness.

Hence, we will be able to compress the $d^{(1)}$-dimensional vectors down to $m$-dimensions. In particular this means we will be able to compress down from $\mathcal{O}(\frac{\log n}{\delta})$ to $\mathcal{O}(\log n)$ dimensions. *However*, note that the key in the operation above is the XOR operator, so we will need to use threshold activation to simulate the XOR operation. The next lemma says that we can transform $\boldsymbol{z}_i^{(1)}$ to $\boldsymbol{z}_i^{(2)}$ using a threshold network containing $\mathcal{O}(\frac{\log^2 n}{\delta})$ neurons and $\mathcal{O}(\frac{\log^2 n}{\delta})$ weights.

**Lemma 4.** *Let $\boldsymbol{z}_i^{(2)}$ be as defined in (2), for all $i \in [n]$. Then, we can construct a threshold network with $3d^{(1)}m$ neurons and $9d^{(1)}m$ weights and biases such that it outputs $\boldsymbol{z}_i^{(1)}$ when the input to it is $\boldsymbol{z}_i^{(2)}$, for all $i \in [n]$.*

Hence, when the input to the overall network is $\boldsymbol{x}_i$, the network constructed up till now will output $\boldsymbol{z}_i^{(2)}$. Similar to the first layer, let $d^{(2)} = m$ denote the dimension of $\boldsymbol{z}_i^{(2)}$. Note that this step uses $\mathcal{O}(\frac{\log^2 n}{\delta})$ neurons and $\mathcal{O}(\frac{\log^2 n}{\delta})$ weights.

**Step 3: Partitioning the dataset into $\widetilde{\mathcal{O}}(\sqrt{n})$ subsets of size at most $\sqrt{n}$ each.** In this step, we partition $\{\boldsymbol{z}_1^{(2)}, \ldots, \boldsymbol{z}_n^{(2)}\}$ into $\mathcal{O}(\sqrt{n} \log n)$ subsets of size at most $\sqrt{n}$ each. The reason why we do this will be apparent in **Step 4**. The key property that we want these subsets to have is that each subset should have a unique prefix such that all the $\boldsymbol{z}_i^{(2)}$ that belong to one subset share that prefix. This will help in detecting which subset a particular $\boldsymbol{z}_i^{(2)}$ belongs to, using threshold networks. Hence, first we will show that such prefixes exist.

**Lemma 5.** *Given $n$ distinct binary vectors $\boldsymbol{z}_1^{(2)}, \ldots, \boldsymbol{z}_n^{(2)}$ of length $d^{(2)} = c \log_2 n$ each (for some constant $c \geq 1$), we can partition them into $\mathcal{O}(\sqrt{n} \log n)$ subsets of size at most $\sqrt{n}$ each, with the important property that each of these subsets will be characterized by unique prefixes, such that all of the elements of one subset will have the same prefix.*

Lemma 5 gives us $K = \mathcal{O}(\sqrt{n} \log n)$ subsets which partition $\{\boldsymbol{z}_1^{(2)}, \ldots, \boldsymbol{z}_n^{(2)}\}$, each with a unique prefix. Let the subsets be $S_1, \ldots, S_K$. Let the binary prefix for $S_i$ be $\boldsymbol{b}_i$, which a length $1 \leq l \leq d^{(2)}$. Then the membership of a vector $\boldsymbol{z}$ can easily be detected by the neuron $s_i : \{0,1\}^{d^{(2)}} \to \{0,1\}$ given by $s_i(\boldsymbol{z}) = \sigma(\sum_{j=1}^{l}(2b_{i,j} - 1)z_j - t)$, where $t$ is the number of 1's in $\boldsymbol{b}_i$. Hence, we can have $K$ neurons, one for each subset, such that $s_i(\boldsymbol{z}) = 1$ if and only if $\boldsymbol{z} \in S_i$. Let $\boldsymbol{s}(\boldsymbol{z})$ represent the

vector formed after concatenating the outputs of all these neurons. Along with these $K$ neurons, we will also have $d^{(2)}$ other neurons that simply copy $\boldsymbol{z}^{(2)}$ over to the next layer. These $d^{(2)}$ neurons can simply have the form $f_i(\boldsymbol{z}) = \sigma(2z_i - 1)$, for $i = 1, \ldots, d^{(2)}$. Overall, if $z_i^{(2)} \in S_j$, then this layer outputs

$$\begin{bmatrix} \boldsymbol{s}(\boldsymbol{z}_i^{(2)}) \\ \boldsymbol{z}_i^{(2)} \end{bmatrix}, \tag{3}$$

as shown in the middle section of Figure 1. Thus, for **Step 3**, we have one layer which consists of $K$ subset indicator neurons, along with $d^{(2)}$ neurons that simply copy $\boldsymbol{z}$ over to the output. Overall, in this step we used one layer with $K + d^{(2)}$ neuron and $(K+1)d^{(2)}$ weights.

**Step 4:** The reason we divided the vectors into subsets and created the indicator neurons is so that the next layer can project the $d^{(2)} = \mathcal{O}(\log n)$ dimensional vectors $\boldsymbol{z}_i^{(2)}$ to $d^{(3)} = \mathcal{O}(\sqrt{n} \log^2 n)$ dimension vectors using the subset indicators. Let $\boldsymbol{z}_i^{(3)}$ be the projection of $\boldsymbol{z}_i^{(2)}$. The way we project will ensure that if we take one vector from each of the $K$ subsets, then these vectors would be linearly independent. This linear independence of $K$ vectors would help in memorizing $K$ samples with just one neuron in the second-to-last layer, which means that we can memorize all the $n$ samples with $\sqrt{n}$ neurons in the second-to-last layer. We explain all of this in detail now.

**Step 4(a): Projecting the vectors up to dimension $\mathcal{O}(\sqrt{n} \log^2 n)$.** In this layer, we will have $K(d^{(2)} + 1)$ neurons partitioned into groups of size $d^{(2)} + 1$ each. As explained in the main paper, the goal would be carry out the transformation shown in Figure 3.

To see how this happens, recall that the previous layer outputs (3). Then, the $j$-th neuron of the $i$-th group in this layer would have the form $g_{i,j} = \sigma(s_i - 1)$ for $j = 1$ and $g_{i,j} = \sigma_{i,j}(s_i + z_j^{(2)} - 2)$, for $j = 2, \ldots, d^{(2)} + 1$. It can be verified that this implements the transformation shown in Figure 3.

This layer will have $K(d^{(2)} + 1)$ neurons and $\leq 3K(d^{(2)} + 1)$ weights and biases. Further, looking at neurons, note that the weights and bias are integers and less than 2.

**Step 4(b): Memorizing using $\sqrt{n}$ neurons.** In this layer, we have $\sqrt{n}$ neurons. The $i$-th neuron memorizes the $i$-th samples from each of the $K$ subsets. Let $\boldsymbol{z}_{i,j}^{(2)}$ denote the $i$-th vector in $S_j$. The $i$-th neuron will have the weight vector given by

$$\boldsymbol{w}_i^{(3)} = \begin{bmatrix} -t_{i,1} + y_{i,1} - 1 \\ 2\boldsymbol{z}_{i,1}^{(2)} - \mathbf{1} \\ -t_{i,2} + y_{i,2} - 1 \\ 2\boldsymbol{z}_{i,2}^{(2)} - \mathbf{1} \\ \vdots \\ -t_{i,d^{(2)}} + y_{i,d^{(2)}} - 1 \\ 2\boldsymbol{z}_{i,d^{(2)}}^{(2)} - \mathbf{1} \end{bmatrix},$$

where $t_{i,j}$ is the number of 1's in $\boldsymbol{z}_{i,j}^{(2)}$, and $y_{i,j}$ is the label of sample $\boldsymbol{x}_{i,j}$ from the dataset that corresponds to $\boldsymbol{z}_{i,j}^{(2)}$. The biases of these neurons are 0.

To see how this works, consider the vector $\boldsymbol{z}_{i,j}^{(2)}$. This is the $i$-th vector in $S_j$. Then, the corresponding $\boldsymbol{z}_{i,j}^{(3)}$ would have support only in the $j$-th chunk of length $d^{(2)} + 1$. Then,

$$\begin{aligned} (\boldsymbol{w}_i^{(3)})^\top \boldsymbol{z}_{i,j}^{(3)} &= -t_{i,j} + y_{i,j} - 1 + (2\boldsymbol{z}_{i,j}^{(2)} - \mathbf{1})^\top \boldsymbol{z}_{i,j}^{(2)} \\ &= -t_{i,j} + y_{i,j} - 1 + 2t_{i,j} - t_{i,j} \\ &= y_{i,j} - 1. \end{aligned}$$

Hence, $\sigma((\boldsymbol{w}_i^{(3)})^\top \boldsymbol{z}_{i,j}^{(3)}) = y_{i,j}$. Further, for any $k \neq i$,

$$(\boldsymbol{w}_i^{(3)})^\top \boldsymbol{z}_{k,j}^{(3)} = -t_{i,j} + y_{i,j} - 1 + (2\boldsymbol{z}_{i,j}^{(2)} - \mathbf{1})^\top \boldsymbol{z}_{k,j}^{(2)}.$$

Note that $(2z_{i,j}^{(2)} - 1)^\top z_{j,j}^{(2)} \le t_{i,j} - 1$ if $z_{i,j}^{(2)} \ne z_{k,j}^{(2)}$. Hence,

$$(w_i^{(3)})^\top z_{k,j}^{(3)} \le y_{i,j} - 2.$$

Hence, $\sigma((w_i^{(3)})^\top z_{k,j}^{(3)}) = 0$.

Thus we see that in this layer, the $i$-th neuron memorizes the $i$-th samples from each subset $\{S_1, \ldots, S_K\}$, and for any other sample, it outputs 0. Since there are at most $\sqrt{n}$ samples in each subset, we need $\sqrt{n}$ neurons in this layer.

Let $[q_1, \ldots, q_{\sqrt{n}}]$ be the concatenated output of this layer. Then, the last layer is a single neuron, $\sigma(2q_1 + \cdots + 2q_{\sqrt{n}} - 1)$, which essentially just sums up the outputs of the previous layer. Since each neuron of the previous layer memorized a subset of the samples, this neuron is able to output the correct label for the entire dataset.

$\square$

## B.1 Proof of helper lemmas for Theorem 1

### B.1.1 Proof of Lemma 1

*Proof.* Consider any two samples from the dataset, $x_i$ and $x_j$. Let $u$ and $v$ be two orthonormal vectors that form a basis of the 2-dimensional space spanned by $x_i$ and $x_j$. Then, any point in the subspace can be written as $z_1 u + z_2 v$, for some scalars $z_1$ and $z_2$. Let $x_i = z_{i,1} u + z_{i,2} v$ and $x_j = z_{j,1} u + z_{j,2} v$.

As per our construction of the hyperplanes, let $\mathbf{w}^\top x = 0$ be any one of the $m$ hyperplanes. Then, we want to find the probability that $(\mathbf{w}^\top x_i)(\mathbf{w}^\top x_j) < 0$. For this, it would be sufficient that $\mathbf{w}^\top x_i > 0$ and $\mathbf{w}^\top x_j < 0$. These are equivalent to $z_{i,1}\mathbf{w}^\top u + z_{i,2}\mathbf{w}^\top v > 0$ and $z_{j,1}\mathbf{w}^\top u + z_{j,2}\mathbf{w}^\top v < 0$. Note that since $\mathbf{w} \sim \mathcal{N}(\mathbf{0}, I_d)$, we have that $\mathbf{w}^\top u \sim \mathcal{N}(0, 1)$ and $\mathbf{w}^\top v \sim \mathcal{N}(0, 1)$, and are independent. For convenience, define the random variables $g_1 := \mathbf{w}^\top u$, $g_2 := \mathbf{w}^\top v$, and $\mathbf{g} := [g_1 \quad g_2]^\top$. Also, define $z_i = [z_{i,1} \quad z_{i,2}]^\top$ and $z_j = [z_{j,1} \quad z_{j,2}]^\top$. Then, we want to bound the probability of event

$$\{\mathbf{g}^\top z_i > 0 \text{ and } \mathbf{g}^\top z_j < 0\}. \tag{4}$$

Note that $\mathbf{g}, z_i$, and $z_j$ are all in 2 dimensions, the angle between $z_i$ and $z_j$ is at least $\delta$. Thus, for (4) to be true, $\mathbf{g}$ should lie in a cone of angle $\delta$ (see Figure 2). Noting that $\mathbf{g}$ is rotationally symmetric, we get that the probability of (4) is exactly $\frac{\delta}{2\pi}$. Hence, the probability that the hyperplane does not separate $z_i$ and $z_j$, is less than $1 - \frac{\delta}{2\pi}$. Since we have $m$ independently sampled hyperplanes, the probability that none of them separates $z_i$ and $z_j$ is less than $(1 - \frac{\delta}{2\pi})^m$. This is the probability that one of the pairs $x_i$ and $x_j$ does not get separated. Since there are a total of $\binom{n}{2}$ pairs of points $x_i$ and $x_j$, then using the union bound, the probability that there exists one pair of points such that the $m$ hyperplanes do not separate them, is less than $\binom{n}{2}(1 - \frac{\delta}{2\pi})^m$. We want this probability to be less than $\epsilon$, that is,

$$\binom{n}{2}\left(1 - \frac{\delta}{2\pi}\right)^m \le \epsilon.$$

Note that $\binom{n}{2} \le n^2$ and $\left(1 - \frac{\delta}{2\pi}\right)^m \le e^{-\frac{\delta}{2\pi}m}$. Hence, it is sufficient for the following to hold

$$n^2 e^{-\frac{\delta}{2\pi}m} \le \epsilon.$$

This gives that $m \ge \frac{2\pi}{\delta}\log\left(\frac{n^2}{\epsilon}\right)$ is sufficient for the lemma to be true. $\square$

### B.1.2 Proof of Lemma 2

This proof is similar to the proof of Lemma 1.

*Proof.* Consider any two samples from the dataset, $x_i$ and $x_j$. Let $u$ and $v$ be two orthonormal vectors that form a basis of the 2-dimensional space spanned by $x_i$ and $x_j$, such that $v \perp (x_i - x_j)$.

If $x_i$ and $x_j$ do not span a 2-dimensional space, then take $u$ to be in the direction of $x_i$, and $v$ can be in any other orthogonal direction. Then, any point in the subspace can be written as $z_1 u + z_2 v$, for some scalars $z_1$ and $z_2$. Let $x_i = z_{i,1} u + z_{i,2} v$ and $x_j = z_{j,1} u + z_{j,2} v$. Note that because $v \perp (x_i - x_j)$, we have that $z_{i,2} = z_{j,2}$, and also $|z_{i,1} - z_{j,1}| = \|x_i - x_j\| \geq \delta$.

As per our construction of the hyperplanes, let $\mathbf{w}^\top x + \mathrm{b} = 0$ be any one of the $m$ hyperplanes. We are interested in the probability that $(\mathbf{w}^\top x_i + \mathrm{b}_i)(\mathbf{w}^\top x_j + \mathrm{b}_j) < 0$, or equivalently

$$(z_{i,1}\mathbf{w}^\top u + z_{i,2}\mathbf{w}^\top v + \mathrm{b})(z_{j,1}\mathbf{w}^\top u + z_{j,2}\mathbf{w}^\top v + \mathrm{b}) < 0. \tag{5}$$

Note that since $\mathbf{w} \sim \mathcal{N}(\mathbf{0}, I_d)$, we have that $\mathbf{w}^\top u \sim \mathcal{N}(0,1)$ and $\mathbf{w}^\top v \sim \mathcal{N}(0,1)$, and are independent. For convenience, define the random variables $\mathrm{g}_1 := \mathbf{w}^\top u$, and $\mathrm{g}_2 := \mathbf{w}^\top v$. Let $(s, t)$ be the coordinates in the 2 dimensional space spanned by $u$ and $v$. Then, note from (5) that we are interested in the behaviour of the hyperplane

$$\mathrm{g}_1 s + \mathrm{g}_2 t + \mathrm{b} = 0, \tag{6}$$

where $\mathrm{g}_1, \mathrm{g}_2$ are the coefficients and $\mathrm{b}$ is the bias. Note that this is just a line in 2-dimensions. In particular, (5) is still equivalent to the event that this line passes in between $x_i$ and $x_j$ in this 2-dimensional space.

Looking at the line (6), its slope is $-\mathrm{g}_1/\mathrm{g}_2$ and its (signed) distance from the origin is $\mathrm{b}/\sqrt{\mathrm{g}_1^2 + \mathrm{g}_2^2}$. Next, note that if we consider the two dimensional Gaussian vector $[\mathrm{g}_2\ \mathrm{g}_1]^\top \sim \mathcal{N}(\mathbf{0}, I_2)$, then thinking in terms of polar coordinates, $\sqrt{\mathrm{g}_1^2 + \mathrm{g}_2^2}$ is its norm and $-\mathrm{g}_1/\mathrm{g}_2$ is its direction. By the spherical symmetry of Gaussians, we know that the norm and its direction are independent. This implies that $\mathrm{b}/\sqrt{\mathrm{g}_1^2 + \mathrm{g}_2^2}$ is independent of $-\mathrm{g}_1/\mathrm{g}_2$, that is the distance of the line from the origin is independent of its slope. Note that $r := \sqrt{2}\mathrm{b}/\sqrt{\mathrm{g}_1^2 + \mathrm{g}_2^2}$ is the (scaled) ratio of a Gaussian random variable with the square-root of a $\chi^2$-random variable, and hence $r$ has a Student's $t$-distribution with 2 degrees of freedom.

Fix the slope $\theta := -\mathrm{g}_1/\mathrm{g}_2$ of the line (6), then we have seen that the signed distance from the origin is the independent random variable $\mathrm{r}/\sqrt{2}$. Then, for the line (6) to pass in between $x_i, x_j$, we will need to have that $\frac{\mathrm{r}}{\sqrt{2}}$ should be between $\left(\mathrm{k}\frac{z_{i,2}+\theta z_{i,1}}{\sqrt{1+\theta^2}}\right)$ and $\left(\mathrm{k}\frac{z_{j,2}+\theta z_{j,1}}{\sqrt{1+\theta^2}}\right)$, where $\mathrm{k} = -1$ or $\mathrm{k} = 1$ depending on the sign of $\theta$. These two exact values do not matter much. What is more important is that since $\|x_i\| \leq 1$ and $\|x_j\| \leq 1$, we know that these values are less than 1; and that

$$\left|\left(\mathrm{k}\frac{z_{i,2} + \theta z_{i,1}}{\sqrt{1+\theta^2}}\right) - \left(\mathrm{k}\frac{z_{j,2} + \theta z_{j,1}}{\sqrt{1+\theta^2}}\right)\right| = \frac{|\theta|}{\sqrt{1+\theta^2}}|z_{j,1} - z_{i,1}|$$
$$\text{(Since } \mathrm{k} \in \{-1, 1\} \text{ and } z_{i,2} = z_{j,2}.)$$
$$\geq \delta\frac{|\theta|}{\sqrt{1+\theta^2}}.$$

In short, once we fix the slope, $\theta$, the probability that the line passes in between the two points is the same as that of a scaled Student's $t$-variable's value being in an interval of length at least $\delta\frac{|\theta|}{\sqrt{1+\theta^2}}$. Note that the interval lies completely in $[-1, 1]$. Because in the interval $[-\sqrt{2}, \sqrt{2}]$, a Student's $t$-variable has its p.d.f. lower bounded by a univeral constant $C$, we get that the probability that the line passes in between the two points is at least $2C\sqrt{2}\delta\frac{|\theta|}{\sqrt{1+\theta^2}}$.

To compute a lower bound on the final probability, we need to unfix the slope $\theta$ and integrate the lower bound we computed above, over the distribution of the slope. Define the polar angle $\phi := \arctan(\theta) = \arctan(-g_1/g_2)$. Note that due to the spherical symmetry of the Gaussian vector $[g_1\ g_2]$, we will have that $\phi$ is uniformly distributed in $[0, 2\pi]$. Hence, the final probability is lower bounded by

$$\frac{1}{2\pi}\int_0^{2\pi} C\sqrt{2}\delta\frac{|\theta|}{\sqrt{1+\theta^2}}\mathrm{d}\phi = \frac{1}{2\pi}C\sqrt{2}\delta\int_0^\pi |\sin(\phi)|\mathrm{d}\phi = C'\delta,$$

for some universal constant $C'$.

What we have shown above, is that the probability that a hyperplane separates $x_i$ and $x_j$ is at least $C'\delta$. Hence, the probability that the hyperplane does not separate $z_i$ and $z_j$, is less than $1 - C'\delta$. Since we have $m$ independently sampled hyperplanes, the probability that none of them separate $z_i$ and $z_j$, is less than $(1 - C'\delta)^m$. This is the probability that one of the pairs $x_i$ and $x_j$ does not get separated. Since there are a total of $\binom{n}{2}$ pairs of points $x_i$ and $x_j$, then by using the union bound, the probability that there exists one pair of points such that the $m$ hyperplanes does not separate them, is less than $\binom{n}{2}(1 - C'\delta)^m$. We want this probability to be less than $\epsilon$, that is,

$$\binom{n}{2}(1 - C'\delta)^m \leq \epsilon.$$

Note that $\binom{n}{2} \leq n^2$ and $(1 - C'\delta)^m \leq e^{-C'\delta m}$. Hence, it is sufficient for the following to hold

$$n^2 e^{-C'\delta m} \leq \epsilon.$$

This gives that $m \geq \frac{1}{C'\delta} \log\left(\frac{n^2}{\epsilon}\right)$ is sufficient for the lemma to be true.

$\square$

### B.1.3  Proof of Lemma 3

*Proof.* Consider two binary vectors $z_i$ and $z_j$ that differ in at least one bit. Without loss of generality, assume that the first bit is one of the bits that they differ in. Let $z_{i,k}$ be the $k$-th element of $z_i$ and similarly let $z_{j,k}$ be the $k$-th element of $z_j$. We can further assume without loss of generality that $z_{i,1} = 0$ and $z_{j,1} = 1$. Then, for any Bernoulli(0.5) vector $\mathbf{b}$, we have

$$\langle z_i, \mathbf{b} \rangle_\oplus = (0 \cdot \mathbf{b}_1) \oplus (z_{i,2} \cdot \mathbf{b}_2) \oplus \cdots \oplus (z_{i,d'} \cdot \mathbf{b}_{d'}),$$
$$\langle z_j, \mathbf{b} \rangle_\oplus = (1 \cdot \mathbf{b}_1) \oplus (z_{j,2} \cdot \mathbf{b}_2) \oplus \cdots \oplus (z_{j,d'} \cdot \mathbf{b}_{d'}),$$

where $\mathbf{b}_k$ is the $k$-th element of $\mathbf{b}$. There can be two cases:

- $(z_{i,2} \cdot \mathbf{b}_2) \oplus \cdots \oplus (z_{i,d'} \cdot \mathbf{b}_{d'}) = (z_{j,2} \cdot \mathbf{b}_2) \oplus \cdots \oplus (z_{j,d'} \cdot \mathbf{b}_{d'})$. Say the probability of this is $p$. Note that $\mathbf{b}_1$ is independent of every other $\mathbf{b}_i$. With probability 0.5, $\mathbf{b}_1 = 1$. In this case,

$$\begin{aligned}
\langle z_i, \mathbf{b} \rangle_\oplus &= (0 \cdot 1) \oplus (z_{i,2} \cdot \mathbf{b}_2) \oplus \cdots \oplus (z_{i,d'} \cdot \mathbf{b}_{d'}) \\
&= 0 \oplus (z_{i,2} \cdot \mathbf{b}_2) \oplus \cdots \oplus (z_{i,d'} \cdot \mathbf{b}_{d'}) \\
&= (z_{i,2} \cdot \mathbf{b}_2) \oplus \cdots \oplus (z_{i,d'} \cdot \mathbf{b}_{d'}) \\
&= \neg(\neg((z_{i,2} \cdot \mathbf{b}_2) \oplus \cdots \oplus (z_{i,d'} \cdot \mathbf{b}_{d'}))) \\
&\qquad \text{(Applying two NOT operations does not affect the value.)} \\
&= \neg(1 \oplus (z_{i,2} \cdot \mathbf{b}_2) \oplus \cdots \oplus (z_{i,d'} \cdot \mathbf{b}_{d'})) \\
&\qquad \text{(XOR with 1 is equivalent to the NOT operation.)} \\
&= \neg((1 \cdot 1) \oplus (z_{i,2} \cdot \mathbf{b}_2) \oplus \cdots \oplus (z_{i,d'} \cdot \mathbf{b}_{d'})) \\
&= \neg\langle z_j, \mathbf{b} \rangle_\oplus.
\end{aligned}$$

  Hence, in this case, with probability at least 0.5, $\langle z_i, \mathbf{b} \rangle_\oplus \neq \langle z_j, \mathbf{b} \rangle_\oplus$.

- $(z_{i,2} \cdot \mathbf{b}_2) \oplus \cdots \oplus (z_{i,d'} \cdot \mathbf{b}_{d'}) = \neg((z_{j,2} \cdot \mathbf{b}_2) \oplus \cdots \oplus (z_{j,d'} \cdot \mathbf{b}_{d'}))$. The probability of this is $1 - p$. Note that $\mathbf{b}_1$ is independent of every other $\mathbf{b}_i$. With probability 0.5, $\mathbf{b}_1 = 0$. In this case,

$$\begin{aligned}
\langle z_i, \mathbf{b} \rangle_\oplus &= (0 \cdot 0) \oplus (z_{i,2} \cdot \mathbf{b}_2) \oplus \cdots \oplus (z_{i,d'} \cdot \mathbf{b}_{d'}) \\
&= 0 \oplus (z_{i,2} \cdot \mathbf{b}_2) \oplus \cdots \oplus (z_{i,d'} \cdot \mathbf{b}_{d'}) \\
&= (z_{i,2} \cdot \mathbf{b}_2) \oplus \cdots \oplus (z_{i,d'} \cdot \mathbf{b}_{d'}) \\
&= \neg((z_{j,2} \cdot \mathbf{b}_2) \oplus \cdots \oplus (z_{j,d'} \cdot \mathbf{b}_{d'})) \\
&= \neg((1 \cdot 0) \oplus (z_{j,2} \cdot \mathbf{b}_2) \oplus \cdots \oplus (z_{j,d'} \cdot \mathbf{b}_{d'})) \\
&= \neg\langle z_j, \mathbf{b} \rangle_\oplus
\end{aligned}$$

  Hence, in this case as well, with probability at least 0.5, $\langle z_i, \mathbf{b} \rangle_\oplus \neq \langle z_j, \mathbf{b} \rangle_\oplus$.

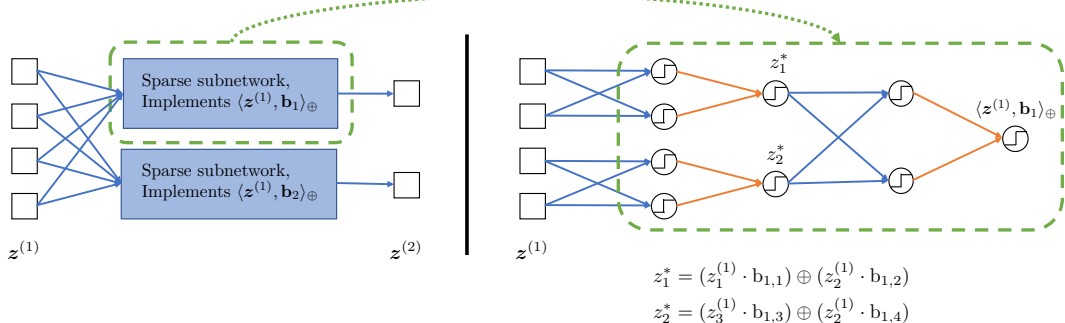

$$z_1^* = (z_1^{(1)} \cdot b_{1,1}) \oplus (z_2^{(1)} \cdot b_{1,2})$$
$$z_2^* = (z_3^{(1)} \cdot b_{1,3}) \oplus (z_2^{(1)} \cdot b_{1,4})$$

Figure 5: **Left:** We can compute each of $\langle \boldsymbol{z}^{(1)}, \mathbf{b}_k \rangle$ using subnetworks shown in blue, that we can put in parallel to output $\boldsymbol{z}^{(2)}$. **Right:** A subnetwork that implements $\langle \boldsymbol{z}^{(1)}, \mathbf{b}_1 \rangle$. This, in turn can be broken down as $\langle \boldsymbol{z}^{(1)}, \mathbf{b}_1 \rangle_\oplus = (z_1^{(1)} \cdot b_{1,1}) \oplus (z_2^{(1)} \cdot b_{1,2}) \oplus \cdots \oplus (z_{d^{(1)}}^{(1)} \cdot b_{1,d^{(1)}})$. Then, the figure on the right shows how the network computes $\langle \boldsymbol{z}^{(1)}, \mathbf{b}_1 \rangle$ using a binary tree structure (see the orange lines in the figure). In the figure, we take $d^{(1)} = 4$. The network first computes $z_1^* = (z_1^{(1)} \cdot b_{1,1}) \oplus (z_2^{(1)} \cdot b_{1,2})$ and $z_2^* = (z_3^{(1)} \cdot b_{1,3}) \oplus (z_4^{(1)} \cdot b_{1,4})$. Then, it computes $z_1^* \oplus z_2^*$, as shown in the figure.

Hence overall, with probability at least $p(0.5)+(1-p)(0.5) = 0.5$, we have that $\langle \boldsymbol{z}_i, \mathbf{b} \rangle_\oplus \neq \langle \boldsymbol{z}_j, \mathbf{b} \rangle_\oplus$. If we take $m$ i.i.d. Bernoulli(0.5) vectors $\mathbf{b}_1, \ldots, \mathbf{b}_m$, then the probability that

$$\forall k \in [m], \langle \boldsymbol{z}_i, \mathbf{b}_k \rangle_\oplus = \langle \boldsymbol{z}_j, \mathbf{b}_k \rangle_\oplus,$$

is less than $(0.5)^m$. Equivalently,

$$\Pr(\boldsymbol{z}_i^{(2)} = \boldsymbol{z}_j^{(2)}) \leq 2^{-m}.$$

To get the bound required in the lemma statement, we simply take a union bound over all the $\binom{n}{2}$ pairs $(\boldsymbol{z}_i^{(2)}, \boldsymbol{z}_j^{(2)})$. Concretely, we want $\Pr(\exists i, j : \boldsymbol{z}_i^{(2)} = \boldsymbol{z}_j^{(2)}) \leq \epsilon$. Hence, we take a union bound:

$$\Pr(\exists i, j : \boldsymbol{z}_i^{(2)} = \boldsymbol{z}_j^{(2)}) \leq \binom{n}{2} \Pr(\boldsymbol{z}_i^{(2)} = \boldsymbol{z}_j^{(2)})$$
$$\leq \binom{n}{2} 2^{-m}.$$

It can be easily verified that $m \geq 3 \log \frac{n}{\epsilon}$ suffices for $\binom{n}{2} 2^{-m} \leq \epsilon$. This completes the proof. $\qquad \square$

### B.1.4 Proof of Lemma 4

*Proof.* We will show that we can compute $\langle \boldsymbol{z}_i^{(1)}, \mathbf{b}_k \rangle_\oplus$ with a network of $3(d^{(1)} - 1)$ neurons and $9(d^{(1)} - 1)$ weights. Then, we can put $m$ such networks in parallel (see Figure 5, Left) to output

$$\boldsymbol{z}_i^{(2)} = \begin{bmatrix} \langle \mathbf{b}_1, \boldsymbol{z}_i \rangle_\oplus \\ \vdots \\ \langle \mathbf{b}_m, \boldsymbol{z}_i \rangle_\oplus \end{bmatrix}.$$

Hence, all we need to do is to prove that we can compute $\langle \boldsymbol{z}_i^{(1)}, \mathbf{b}_k \rangle_\oplus$ with a network of $3(d^{(1)} - 1)$ neurons and $9(d^{(1)} - 1)$ weights. We will do this in a hierarchical fashion as shown in Figure 5, Right. We begin by expanding $\langle \boldsymbol{z}_i^{(1)}, \mathbf{b}_k \rangle_\oplus$:

$$\langle \boldsymbol{z}_i^{(1)}, \mathbf{b}_k \rangle_\oplus = (z_{i,1}^{(1)} \cdot b_{k,1}) \oplus (z_{i,2}^{(1)} \cdot b_{k,2}) \oplus \cdots \oplus (z_{i,d^{(1)}}^{(1)} \cdot b_{k,d^{(1)}}).$$

Note that

$$(z_{i,1}^{(1)} \cdot b_{k,1}) \oplus (z_{i,2}^{(1)} \cdot b_{k,2}) = \sigma(\sigma((z_{i,1}^{(1)} \cdot b_{k,1}) - (z_{i,2}^{(1)} \cdot b_{k,2}) - 1) + \sigma(-(z_{i,1}^{(1)} \cdot b_{k,1}) + (z_{i,2}^{(1)} \cdot b_{k,2}) - 1))),$$

which uses 3 neurons, each with 2 weights and a bias. Similarly, we can compute $(z_{i,3}^{(1)} \cdot \mathbf{b}_{k,3}) \oplus (z_{i,4}^{(1)} \cdot \mathbf{b}_{k,4})$, $(z_{i,5}^{(1)} \cdot \mathbf{b}_{k,5}) \oplus (z_{i,6}^{(1)} \cdot \mathbf{b}_{k,6})$, $\ldots$, $(z_{i,d^{(1)}-1}^{(1)} \cdot \mathbf{b}_{k,d^{(1)}-1}) \oplus (z_{i,d^{(1)}}^{(1)} \cdot \mathbf{b}_{k,d^{(1)}})$. Once we have computed these, we can compute XOR's of four terms similarly, to get terms like $(z_{i,1}^{(1)} \cdot \mathbf{b}_{k,1}) \oplus (z_{i,2}^{(1)} \cdot \mathbf{b}_{k,2}) \oplus$. We keep applying this hierarchically in a binary tree form to get $\langle \mathbf{z}_i, \mathbf{b}_k \rangle_{\oplus} \oplus (z_{i,3}^{(1)} \cdot \mathbf{b}_{k,3}) \oplus (z_{i,4}^{(1)} \cdot \mathbf{b}_{k,4})$ (see Figure 5, Right). The tree will consist of $d^{(1)} - 1$ XOR operations, each represented by a node. For each of these operations, we saw above that we need 3 neurons, each with 2 weights and a bias. This gives that we need $3(d^{(1)} - 1)$ neurons and $9(d^{(1)} - 1)$ weights and biases to compute $\langle \mathbf{z}_i^{(1)}, \mathbf{b}_k \rangle_{\oplus}$.

$\square$

### B.1.5 Proof of Lemma 5

*Proof.* Consider a binary tree of depth $d^{(2)} + 1$, were the left child edge of each node is labeled 0, and the right one is labeled 1. There will be $2^{d^{(2)}}$ leaves, each representing one unique binary vector of length $d^{(2)}$. The vector associated with a leaf would be the same as the sequence of edges of the path from the root to that leaf. Hence, $\{\mathbf{z}_1^{(2)}, \ldots, \mathbf{z}_n^{(2)}\}$ will form a subset of the $2^{d^{(2)}}$ leaves. Remove all other leaves from the tree, so that we only have $\{\mathbf{z}_1^{(2)}, \ldots, \mathbf{z}_n^{(2)}\}$ as the leaves in our tree. Hence, now our tree only has $n$ leaves. Also remove all the nodes that are not the ancestor of any leaves.

The way we select subsets is simple: Each node is a potential subset. Each node represents the subset of $\{\mathbf{z}_1^{(2)}, \ldots, \mathbf{z}_n^{(2)}\}$ that belong in the subtree at that node. Each node is also associated with a unique prefix: the sequence of edges of the path from the root to that node. Denote by $\mathcal{S}$ the set of nodes (or equivalently subsets) that we select. We include a node in $\mathcal{S}$, if

1. the number of leaves in the subtree at that node is at most $\sqrt{n}$,

2. none of its ancestor nodes satisfy condition 1.

Condition 1 will ensure that the subsets formed in this way will have size at most $\sqrt{n}$. Condition 2 above ensures that the subsets do not overlap and consequently the prefixes of the selected nodes are unique. Hence, all that is left to do is prove that $|\mathcal{S}| = \mathcal{O}(\sqrt{n} \log n)$.

Define the height of a node as $(d^{(2)} - $ its depth$)$. Thus, the root node has height $d^{(2)}$ and the leaves have height 0. We will prove that for any node at height $h$, if any of its descendant nodes is chosen, then

$$\frac{\# \text{ leaves in the subtree at that node}}{\# \text{ descendant nodes included in } \mathcal{S}} \geq \frac{\sqrt{n}}{h}. \tag{7}$$

At root node, $h = d^{(2)} = c \log n$, and # leaves $= n$. Then, rearranging the inequality (7), we get that at the root node the number of descendant nodes included in $\mathcal{S}$ are less than $\sqrt{n}(c \log n)$. Thus, the total number of subsets is $\mathcal{O}(\sqrt{n} \log n)$. Hence, all we need to do now, is to prove (7). For this, we will use induction on $h$.

The smallest $h$ for which the inequality is applicable, will be $h = \lceil \log_2 \sqrt{n} \rceil +$. This is because the first height at which a node can be selected is $\lceil \log_2 \sqrt{n} \rceil$. Note that no descendants of a node at height $h = \lceil \log_2 \sqrt{n} \rceil + 1$, other than its children, can be included in $\mathcal{S}$. Further, since one of its children is included in $\mathcal{S}$, this node itself is not included in $\mathcal{S}$. Hence, the number of leaves in the subtree at this node is at least $\sqrt{n} + 1$. Hence for this node,

$$\frac{\# \text{ leaves in the subtree formed at that node}}{\# \text{ descendant nodes in } \mathcal{S}} \geq \frac{\sqrt{n} + 1}{2} \geq \frac{\sqrt{n}}{h}.$$

Thus, the base case is proved. We move on to the inductive case now: $h > \lceil \log_2 \sqrt{n} \rceil + 1$. There can be the following cases for a node $s$ at height $h$.

- $s$ has only one child. This child cannot be in $\mathcal{S}$, since Condition 2 would be violated. Thus, it can only be the case that this child has further descendants that are in $\mathcal{S}$. Then, by the induction hypothesis, (7) is satisfied for the child, and hence it is also satisfied for $s$.

- Both of its children have descendants that are in $\mathcal{S}$. Let $l_l$ be the (# leaves in the subtree formed at the left child of $s$) and $d_l$ be the (# descendant nodes of the left child in $\mathcal{S}$). Similarly, we define $l_r$ and $d_r$ for the right child. Then, we know by the induction hypothesis that

$$\frac{l_l}{d_l} \geq \frac{\sqrt{n}}{h-1}, \frac{l_r}{d_r} \geq \frac{\sqrt{n}}{h-1}.$$

Hence, for the node $s$,

$$\frac{\text{\# leaves in the subtree formed at } s}{\text{\# descendant nodes in } \mathcal{S}} = \frac{l_l + l_r}{d_l + d_r}$$

$$\geq \frac{\frac{\sqrt{n}}{h-1}(d_l + d_r)}{d_l + d_r}$$

$$\geq \frac{\sqrt{n}}{h}.$$

- One of its children is included in $\mathcal{S}$ and the other node has at least one of its descendants included in $\mathcal{S}$. Let the left child be the one that is included in $\mathcal{S}$. Let $l_r$ be the (# leaves in the subtree formed at the right child of $s$) and $d_r$ be the (# descendant nodes of the right child included in $\mathcal{S}$). Then, by inductive hypothesis,

$$\frac{l_r}{d_r} \geq \frac{\sqrt{n}}{h-1}.$$

We also know that since the right child wasn't included in $\mathcal{S}$, $l_r > \sqrt{n}$. Note that the number of leaves under $s$ is at least $l_r$ and the number of its descendants that are in $\mathcal{S}$ is $d_r + 1$, including the left child. Then,

$$\frac{\text{\# leaves in the subtree formed at } s}{\text{\# descendant nodes in } \mathcal{S}} \geq \frac{l_r}{1 + d_r}$$

$$= \frac{1}{\frac{1}{l_r} + \frac{d_r}{l_r}}$$

$$\geq \frac{1}{\frac{1}{l_r} + \frac{h-1}{\sqrt{n}}}$$

$$= \frac{\sqrt{n}}{\frac{\sqrt{n}}{l_r} + h - 1}$$

$$> \frac{\sqrt{n}}{h}.$$

- Both the children of the node are in $\mathcal{S}$. In this case, since the node itself wasn't chosen, the number of leaves in the subtree is at least $\sqrt{n} + 1$, and the number of descendant nodes in $\mathcal{S}$ is 2. Hence,

$$\frac{\text{\# leaves in the subtree formed by that node}}{\text{\# descendant nodes in } \mathcal{S}} \geq \frac{\sqrt{n}+1}{2} \geq \frac{\sqrt{n}}{h}.$$

This proves the inductive case and completes the proof of lemma. □

## B.2 Constructing a shallow network for memorization

The network constructed in this proof has depth $O(\log \frac{\log n}{\delta})$, which is due to the sparse sub-network which compresses the $\mathcal{O}(\frac{\log n}{\delta})$ sized binary representation down to $\mathcal{O}(\log n)$ sized

binary representation. Removing this subnetwork will bring the depth of the resulting network down to 4 hidden layers. However, doing this will also increase the number of neurons needed from $\mathcal{O}(\frac{\log^2 n}{\delta} + \sqrt{n}\log^2 n)$ to $\mathcal{O}(\frac{\sqrt{n}\log^2 n}{\delta})$, and the number of weights needed from $\mathcal{O}(\frac{(d+\log n)\log n}{\delta} + n\log^2 n)$ to $\mathcal{O}(\frac{(d+n)\log^2 n}{\delta} + \frac{\sqrt{n}\log^2 n}{\delta^2})$.

### B.3 Comments on generalization

We remark here that the construction presented in this section does not imply any results on the generalization capacity of the network. However, note that in essence, the first layer partitions the space into convex polytopes and the rest of the network assigns labels to these polytopes. Making the first layer wider would result in a finer partition of the space, into smaller polytopes. Each point within a polytope would have the same binary representation after the first layer, and hence the rest of the network can assign different labels to different polytopes. Finally, if the first layer is wide enough, we can have a very fine partition of the space, and approximate the nearest neighbor classifier. This can lead to guarantees on generalization. Exploring this would be an interesting direction of future research.

## C  Proof of Theorem 2 and Theorem 5

*Proof.* We want to construct an example where there are $n$ points on the unit sphere $\mathcal{S}^{d-1}$, which are separated by a distance at least $\delta$, such that the minimum number of hyperplanes needed for separating each pair is $\Omega(\log(n)/\delta)$ and for separating pairs with opposite labels is $\Omega(1/\delta)$.

Before we start the proof, let us define some notation. We denote by $\mathcal{S}^{d-1} := \{x : \|x\| = 1\}$, the unit sphere in $d$-dimensions centered at the origin. We can also have lower dimensional spheres in the same space. For example, we can have circles in a 3-dimensional space. We denote by $\mathcal{S}_{\boldsymbol{c}}^{d-2}$, the $d-1$ dimensional unit sphere, centered at $\boldsymbol{c}$ and which lies on the hyperplane orthogonal to the vector $\boldsymbol{c}$. For example, in 3-dimensions, let $\boldsymbol{c} = (1, 1, 1)$. Let $\mathcal{H}_{\boldsymbol{c}}$ be the hyperplane that passes through $\boldsymbol{c}$ and is orthogonal to the vector connecting the origin to $\boldsymbol{c}$. On this hyperplane, we can draw a unit circle, centered at $(1, 1, 1)$, which is what we denote by $\mathcal{S}_{(1,1,1)}^{2}$ in our notation. Further, $r\mathcal{S}^{d-1}$ and $r\mathcal{S}_{\boldsymbol{c}}^{d-2}$ will denote the corresponding hyperspheres, but with radius $r$ instead of 1. We denote the cluster centered at $\boldsymbol{c}$ by $\mathcal{Q}_{\boldsymbol{c}}$. We will use $C_1, C_2, \dots$ to denote universal constants.

Now, we are ready to start the construction. We will create $\sqrt{n}$ clusters of points on the sphere, each of size $\sqrt{n}$.

First, the centers of the $\sqrt{n}$ clusters are sampled uniformly from $(\sqrt{1-\delta})\mathcal{S}^{d-1}$. Then, in each cluster, $\sqrt{n}$ points are sampled i.i.d. from $\sqrt{\delta}\mathcal{S}_{\boldsymbol{c}}^{d-2}$, where $\boldsymbol{c}$ is the center of the cluster. This way, each point has norm $\sqrt{(\sqrt{1-\delta})^2 + (\sqrt{\delta})^2} = 1$, that is, each point indeed lies on $\mathcal{S}^d$. Of the $\sqrt{n}$ points in each cluster, we label half of them as 0 and the other half as 1.

Then, roughly speaking, the summary of the proof is as follows:

- Any hyperplane can only 'effectively' separate points within a cluster if it passes close to the center of the cluster. The distance would roughly need to be $\widetilde{O}(\sqrt{\delta}/\sqrt{d})$.

- A hyperplane can be $\widetilde{O}(\sqrt{\delta}/\sqrt{d})$ close to the centers of only $\widetilde{O}(\sqrt{\delta})$ fraction of the clusters.

- Any cluster needs at least

  - $\Omega(\log_2 \sqrt{n}) = \Omega(\log n)$ hyperplanes to separate each pair of points within the cluster (for Theorem 5) or
  - at least 1 hyperplane to separate all the opposite label pairs (for Theorem 2).

- Finally, the core inequality that we use is that

$$(\# \text{ hyperplanes needed}) \times (\# \text{ clusters a hyperplane can 'effectively' separate})$$
$$\geq (\# \text{ clusters}) \times (\# \text{ hyperplanes needed per cluster}). \quad (8)$$

Substituting the bounds we got above into this inequality, we get that we need at least $\Omega(\log(n)/\sqrt{\delta})$ hyperplanes.

Now we describe the construction in detail. Note that we only need to prove the existence of *one* such construction. Thus, we will use the probabilistic method.

We define four events:

$\mathcal{E}_1$ : Two points from different clusters have distance less than $\delta$.

$\mathcal{E}_2$ : The points within a cluster have distance less than $\delta$.

$\mathcal{E}_3$ : A hyperplane at distance $\Omega(\sqrt{\frac{\delta \log t}{d}})$ from the center of a cluster separates $\Omega(n/t)$ pairs of points in a cluster, for $t \geq 4$.

$\mathcal{E}_4$ : A hyperplane passes within distance $O(\sqrt{\frac{\delta \log t}{d}})$ of the centers of $\Omega(\sqrt{\delta n \log t})$ clusters.

We will prove that the probability of each of these events is at most $1/5$. Hence, a union bound shows that there exists an event where none of these events happen. We will prove that such an event gives the required lower bounds.

**Step 1: Bound the probability of $\mathcal{E}_1$.** Let $c_1, \ldots, c_{\sqrt{n}}$ be $\sqrt{n}$ cluster centers sampled uniformly from $(\sqrt{1-\delta})\mathcal{S}^{d-1}$, representing the centers of the clusters. Then there exists a universal constant $C$, such that, as long as $n < \left(\frac{C}{\sqrt{\delta}}\right)^{d-1}$, we show that the event that two points from different clusters have distance less than $\delta$ will have probability less than $1/5$.

To see how this is true, consider any two clusters, with centers $c_i$ and $c_j$. Then, as long as $\|c_i - c_j\| \geq 2\sqrt{\delta} + \delta$, we get by triangle inequality that two points from these two clusters are at least $\delta$ distance apart. Since $\sqrt{\delta} \geq \delta$, we know that if we ensure that all cluster centers are at least $3\sqrt{\delta}$ apart, then the above still holds.

Hence, we need to show that if we sample $\sqrt{n}$ points (cluster centers) on a sphere of radius $\sqrt{1-\delta}$, and $n < \frac{1}{5}\left(\frac{C}{\sqrt{\delta}}\right)^{d-1}$, then they are at least $3\sqrt{\delta}$ apart with probability at least $4/5$. For this, we need to show that the probability of any pair of points $c_i$ and $c_j$ being within $3\sqrt{\delta}$ distance is less than $\left(\frac{\sqrt{\delta}}{C}\right)^{d-1}$. Taking a union bound over all the $\binom{\sqrt{n}}{2}$ pairs and substituting $n < \left(\frac{C}{\sqrt{\delta}}\right)^{d-1}$ will give that the probability of any such pair existing is less than $1/5$.

Thus, all we need to do is to prove that the probability of any pair of points being within $3\sqrt{\delta}$ distance is less than $\left(\frac{\sqrt{\delta}}{C}\right)^{d-1}$. We prove it below.

Fix the point $c_i$. Then, the probability of any other cluster center $c_j$ being within distance $3\sqrt{\delta}$ of $c_i$ is the same as the ratio of the surface area of the cap of radius $3\sqrt{\delta}$ to the surface area of $\sqrt{1-\delta}\mathcal{S}^{d-1}$. Consider packing the surface of $\sqrt{1-\delta}\mathcal{S}^{d-1}$ with caps of radius $3\sqrt{\delta}$, and let $\mathcal{A}_{3\sqrt{\delta}}$ denote such a cap. Let the number of caps required for the packing be $\mathcal{P}'$. Then,

$$\text{Area}(3\sqrt{\delta} \text{ cap}) \times \mathcal{P}' \leq \text{Area}(\sqrt{1-\delta}\mathcal{S}^{d-1})$$

$$\implies \frac{\text{Area}(3\sqrt{\delta} \text{ cap})}{\text{Area}(\sqrt{1-\delta}\mathcal{S}^{d-1})} \leq \frac{1}{\mathcal{P}'}.$$

Further, note that by rescaling $|\mathcal{P}'|$ is equal to $|\mathcal{P}_{3\sqrt{\delta}/\sqrt{1-\delta}}|$, where $\mathcal{P}_{\delta'}$ is the $\delta'$ packing of $\mathcal{S}^{d-1}$. Finally, we use the following lemma (proved later in the appendix) to conclude that the probability of any pair of points being within $3\sqrt{\delta}$ distance is less than $\left(\frac{\sqrt{\delta}}{C}\right)^{d-1}$, for some universal constant $C$.

**Lemma 6.** *Let $\mathcal{C}_\delta$ and $\mathcal{P}_\delta$ denote the covering and packing numbers of $\mathcal{S}^{d-1}$, respectively. Then,*

$$\left(\frac{1}{4\delta}\right)^{d-1} \leq \mathcal{C}_\delta \leq \mathcal{P}_\delta \leq \left(\frac{2}{\delta}\right)^d.$$

**Step 2: Bound the probability of $\mathcal{E}_2$.** Consider any one cluster center $c$. Then, the sphere $\sqrt{\delta}\mathcal{S}_c^{d-2}$ is just a $d-2$ dimensional sphere. We create this cluster by sampling points uniformly on $\mathcal{S}_c^{d-2}$. This is similar to what we did in previous step and hence we can apply exactly the same steps as above to get that as long as $n < \left(\frac{C}{\sqrt{\delta}}\right)^{d-2}$, we have that the event that two points within some cluster have distance less than $\delta$, will have probability less than $1/5$. We denote this cluster by $\mathcal{Q}_c$. Then, $\mathcal{Q}_c$ lies on the sphere $\sqrt{\delta}\mathcal{S}_c^{d-2}$, which in turn lies on the hyperplane $\mathcal{H}_c$.

We need to create such clusters around each of the other cluster centers. A trick that we employ is that we use the same cluster $\mathcal{Q}_c$ and 'paste' it at every other cluster center, after rotating appropriately to make the cluster lie on the hyperplane orthogonal to the corresponding cluster center. This is helpful because we would not have to take a union bound of the probability calculation done above for every cluster. We formally describe the procedure below.

Let $c'$ be another cluster center. Then, there will be a corresponding sphere $\sqrt{\delta}\mathcal{S}_{c'}^{d-2}$, which lies on the hyperplane $\mathcal{H}_{c'}$. Let $T_{c'}$ be any linear transformation such that $T(\sqrt{\delta}\mathcal{S}_c^{d-2}) = \sqrt{\delta}\mathcal{S}_{c'}^{d-2}$. Then, we create the cluster $\mathcal{Q}_{c'}$ for the cluster center $c'$ as $\mathcal{Q}_{c'} = T(\mathcal{Q}_c)$.

**Step 3: Bound the probability of $\mathcal{E}_3$.** Any hyperplane $\mathcal{H}_2$ (distinct from $\mathcal{H}_c$) that intersects $\mathcal{H}_c$ creates a $d-1$ dimensional hyperplane, say $\mathcal{H}_2'$. We want to show that if $\mathcal{H}_2$ lies at distance $\Omega(\sqrt{\frac{\delta \log t}{d}})$ from a cluster center $c$, then the probability that it separates $\geq n/t$ pairs of points in the cluster is less than $1/5$. We will actually show that if $\mathcal{H}_2'$ lies at distance $\Omega(\sqrt{\frac{\delta \log t}{d}})$ from $c$, then the probability that it separates $\Omega(n/t)$ pairs of points in the cluster is less than $1/5$, which is a stronger statement because the distance of $c$ from $\mathcal{H}_2$ is smaller than the distance of $c$ from $\mathcal{H}_2'$. We do not consider hyperplanes parallel to $\mathcal{H}_c$ because such hyperplanes do not separate any points in $\mathcal{Q}_c$. We will work in the $d-1$ dimensional subspace spanned by $\mathcal{H}_c$, with $c$ as the origin.

Take any one direction in this space. The probability measure of $\sqrt{\delta}\mathcal{S}_c^{d-2}$ at distance $\geq 2\sqrt{\frac{\delta \log t}{d-1}}$ in this direction is at most $e^{-2\log t}$ (Ball et al., 1997, Lemma 2.2). Thus, the probability that a point in the cluster lies at a distance more than $2\sqrt{\frac{\delta \log t}{d-1}}$ is less than $t^{-2}$. We can now apply Binomial tail probability bounds to get a probability bound on the event

$$E : \{\text{more than } \frac{1}{t} \text{ fraction of the } \sqrt{n} \text{ points lie at a distance more than } 2\sqrt{\frac{\delta \log t}{d-1}} \text{ in one fixed direction}\}.$$

Let $D(\cdot\|\cdot)$ denote the KL-divergence. Then, the Chernoff bound on the tail probability of a Binomial random variable (Arratia & Gordon, 1989) gives the following bound

$$\begin{aligned}
\Pr(E) &\leq e^{-\sqrt{n}D(\frac{1}{t}\|\frac{1}{t^2})} \\
&= \exp\left(-\sqrt{n}\left(\frac{1}{t}\log t + \left(1-\frac{1}{t}\right)\log\frac{1-\frac{1}{t}}{1-\frac{1}{t^2}}\right)\right) \\
&= \exp\left(-\sqrt{n}\left(\frac{1}{t}\log t + \left(1-\frac{1}{t}\right)\log\frac{t}{1+t}\right)\right) \\
&= \exp\left(-\sqrt{n}\left(\log t - \log(1+t) + \frac{1}{t}\log(1+t+t^2)\right)\right) \\
&= \exp\left(-\sqrt{n}\left(\log\left(\frac{t}{1+t}\right) + \frac{1}{t}\log(1+t)\right)\right) \\
&= \exp\left(-\sqrt{n}\left(\log\left(1-\frac{1}{1+t}\right) + \frac{1}{t}\log(1+t)\right)\right)
\end{aligned}$$

$$\leq \exp\left(-\sqrt{n}\left(-\frac{1}{2(1+t)} + \frac{1}{t}\log(1+t)\right)\right) \qquad \text{(Since } \tfrac{1}{1+t} \leq 1\text{)}$$

$$\leq \exp\left(\frac{-\sqrt{n}}{2t}\log(1+t)\right) \qquad \text{(Since } t \geq 2\text{)}$$

$$\leq \frac{1}{t^{\sqrt{n}/2t}}.$$

Recall that the above result is just for one fixed direction, whereas we want to ensure that no more than $1/t$ fraction of points lie at distance $\Omega(\sqrt{\delta \log t/(d-1)})$ in *any* direction. For this, we need an $\epsilon$-net argument[5] over all the directions. Note that a direction is essentially just a point on the unit sphere and hence, we need to make an $\epsilon$-net on $\mathcal{S}^{d-2}$. We claim that for $\epsilon = \sqrt{\frac{\log t}{d-1}}$, if we prove that no more than $1/t$ fraction of points lie at distance $2\sqrt{\delta \log t/(d-1)}$ in any direction in the $\epsilon$-net, then no more than $1/t$ fraction of points lie at distance $4\sqrt{\delta \log t/(d-1)}$ in any direction. We will prove this claim shortly. Recall that an $\epsilon$-net is the same as a covering and hence we can use Lemma 6 to get an upper bound on the size of the $\epsilon$-net (Note that the radius of the sphere here is $\sqrt{\delta}$, so we need to rescale appropriately). Taking a union bound over the $\epsilon$-net gives the probability

$$p := \left(4\sqrt{\frac{d-1}{\log t}}\right)^{d-1} \frac{1}{t^{\sqrt{n}/2t}}. \tag{9}$$

Thus, any hyperplane that lies at distance more than $4\sqrt{\frac{\delta \log t}{d-1}}$ from the center of a cluster has less than $\sqrt{n}/t$ points on one side of it, with probability at least $1-p$. Note that if a hyperplane has less than $\sqrt{n}/t$ points on one side, then it can only separate $\leq \frac{\sqrt{n}}{t}(\sqrt{n} - \frac{\sqrt{n}}{t}) \leq \frac{n}{t}$ pairs of points, which is what we want to prove in this step. Hence, we want to show that $p < 1/5$. From (9), we can see that this will be true as long as $\sqrt{n} \geq C_3 dt \log d$, where $C_3$ is a large enough universal constant.

To finish this step, we need to prove that an $\epsilon$-net would ensure that in every direction, no more than $1/t$ fraction of points lie beyond $4\sqrt{\frac{\delta \log t}{d-1}}$. For any direction (or equivalently, a unit vector) $\boldsymbol{u}$, there is a direction $\boldsymbol{u}_e$ in our epsilon net such that $\|\boldsymbol{u} - \boldsymbol{u}_e\| \leq 2\sqrt{\frac{\log t}{d-1}}$. Consider the region of $\sqrt{\delta}\mathcal{S}_{\boldsymbol{c}}^{d-2}$ consisting of points that lie at distance $4\sqrt{\frac{\delta \log t}{d-1}}$ from the origin (*i.e.*, $\boldsymbol{c}$) in the direction of $\boldsymbol{u}$. Let $\boldsymbol{x}$ be any point in the region. Then,

$$\langle \boldsymbol{x}, \boldsymbol{u}_e \rangle = \langle \boldsymbol{x}, \boldsymbol{u} \rangle + \langle \boldsymbol{x}, \boldsymbol{u}_e - \boldsymbol{u} \rangle$$

$$\geq 4\sqrt{\frac{\delta \log t}{d-1}} + \langle \boldsymbol{x}, \boldsymbol{u}_e - \boldsymbol{u} \rangle$$

$$\geq 4\sqrt{\frac{\delta \log t}{d-1}} - \|\boldsymbol{x}\|\|\boldsymbol{u}_e - \boldsymbol{u}\|$$

$$\geq 4\sqrt{\frac{\delta \log t}{d-1}} - 2\sqrt{\frac{\delta \log t}{d-1}}$$

$$= 2\sqrt{\frac{\delta \log t}{d-1}}.$$

Hence, any point in such a region lies at distance $\geq 2\sqrt{\frac{\delta \log t}{d-1}}$ in the direction $\boldsymbol{u}_e$.

**Step 4: Bound the probability of $\mathcal{E}_4$.** This is similar to the previous step. We need to show that there are only $\mathcal{O}(\sqrt{\delta n \log t})$ cluster centers within distance $\mathcal{O}(\sqrt{\delta \log t/d})$ of any hyperplane.

Take any fixed hyperplane. Then, the probability measure of the sphere $\sqrt{1-\delta}\mathcal{S}^{d-1}$ within distance $12\sqrt{\delta \log t}/\sqrt{d}$ of the hyperplane is less than $C_5\sqrt{\delta \log t}$, for some universal constant $C_5$. Using

---

[5]An $\epsilon$-net $\mathcal{E}$ of a set $\mathcal{B}$ is a set of points from $\mathcal{B}$ such that any point in $\mathcal{B}$ is at distance at most $\epsilon$ from some point in $\mathcal{E}$.

this, we get that the probability that any cluster center lies within distance $12\sqrt{\delta \log t}/\sqrt{d}$ of the hyperplane is less than $C_5\sqrt{\delta \log t}$. Similar to the previous step, we can now use Chernoff bounds to get the probability that more than $2C_5\sqrt{\delta \log t}$ fraction of cluster centers lie at a distance less than $12\sqrt{\delta \log(t)/d}$ is less than $e^{-C_5\sqrt{n\delta \log t}}$.

Recall that this is for a fixed hyperplane, whereas we want to give a probability bound for all hyperplanes. Hence, similar to the previous step, we take a union bound over an $\epsilon$-net of hyperplanes. We claim that if $\epsilon = 4\sqrt{\delta \log(t)/d}$, then the $\epsilon$-net argument will ensure that for any hyperplane, the probability that more than $2C_5\sqrt{\delta \log t}$ fraction of cluster centers lie at a distance less than $4\sqrt{\delta \log(t)/d}$ is less than $e^{-C_5\sqrt{n\delta \log t}}$. We will prove this shortly. We will see that the size of the net will be $\frac{1}{\epsilon}$ times the the covering number of $\mathcal{S}^{d-1}$. Taking a union bound over the $\epsilon$-net gives the probability

$$p' := \frac{1}{\epsilon}\left(C_6\sqrt{\frac{d}{\delta \log t}}\right)^{d-1} e^{-C_5\sqrt{n\delta \log t}}.$$

$p'$ will be less than $1/5$ when $\sqrt{n} > C_7\frac{d}{\sqrt{\delta}}\log\frac{d}{\delta}$.

Next, we prove that if we create an $\epsilon$-net of hyperplanes, then that would indeed be sufficient to ensure that for any hyperplane, the probability that more than $2C_5\sqrt{\delta \log t}$ fraction of cluster centers lie at a distance less than $4\sqrt{\delta \log(t)/d}$ is less than $e^{-C_5\sqrt{n\delta \log t}}$. A hyperplane can be represented by $h(\boldsymbol{x}) = \boldsymbol{u}^\top\boldsymbol{x} + b$, where $\boldsymbol{u}$ is a unit vector. Similar to previous step, create an $\epsilon$-net over the unit vectors $\boldsymbol{u}$. We also create an $\epsilon$-net over $b$. However, since we are only interested in hyperplanes that intersect the sphere, we restrict $b \in [0, 1]$, and hence the epsilon net over $b$ would be of size $1/\epsilon$. We know that there is a hyperplane $h_e(\boldsymbol{x}) = \boldsymbol{w}_e^\top\boldsymbol{x} + b_e$ in our $\epsilon$-net such that $\|\boldsymbol{w}_e - \boldsymbol{w}\| \leq \epsilon$ and $|b - b_e| \leq \epsilon$. Consider the region of $\sqrt{1-\delta}\mathcal{S}^{d-1}$ within distance $4\sqrt{\delta \log(t)/d}$ of $h$. Hence, for any point $\boldsymbol{x}$ in the region,

$$\begin{aligned}
\boldsymbol{w}_e^\top\boldsymbol{x} + b_e &= (\boldsymbol{w}^\top\boldsymbol{x} + b) + (\boldsymbol{w}_e - \boldsymbol{w})^\top\boldsymbol{x} + (b_e - b) \\
&\leq (\boldsymbol{w}^\top\boldsymbol{x} + b) + (\boldsymbol{w}_e - \boldsymbol{w})^\top\boldsymbol{x} + \epsilon \\
&\leq (\boldsymbol{w}^\top\boldsymbol{x} + b) + \|\boldsymbol{w}_e - \boldsymbol{w}\|\|\boldsymbol{x}\| + \epsilon \\
&\leq (\boldsymbol{w}^\top\boldsymbol{x} + b) + \epsilon + \epsilon \\
&\leq 12\sqrt{\delta \log(t)/d}.
\end{aligned}$$

Hence, any such point lies within distance $12\sqrt{\delta \log(t)/d}$.

**Step 5: Using the core inequality.** We showed in the steps above that with probability at least $1/5$, none of the events $\mathcal{E}_1, \mathcal{E}_1, \mathcal{E}_1, \mathcal{E}_4$ happen. We will work in such an event, *i.e.,* the event when none of $\mathcal{E}_1, \mathcal{E}_1, \mathcal{E}_1, \mathcal{E}_4$ happen.

Let $h$ denote the minimum number of hyperplanes needed to separate the dataset that we have constructed. We set $t = 8h$. Then, we showed in Step 3 that any hyperplane at distance more than $4\sqrt{\delta \log(t)/(d-1)}$ from a cluster center can only separate $n/t = n/8h$ pairs of points, whereas there are a total of $n/4$ pairs of opposite label points in each cluster. Hence, we need at least one hyperplane to pass within distance $4\sqrt{\delta \log(t)/(d-1)}$ of the cluster center, otherwise there would be at least $n/8$ pairs of opposite label points that would not be separated. On the other hand, we showed in Step 4 that a hyperplane can pass within distance $4\sqrt{\delta \log(t)/(d-1)}$ of only $2C_5\sqrt{\delta \log t}$ fraction of the cluster centers. Now, we are ready to use the core inequality (8) to get that

$$h \geq \frac{\sqrt{n} \times 1}{2C_5\sqrt{\delta \log t}\sqrt{n}}.$$

This gives that $h = \Omega(1/(\sqrt{\delta}\log(1/\delta)))$.

For the case when we need to separate every pair of points (Theorem 5), the only difference in the inequality above is that we would need at least $\log_2(n - \frac{n}{8})$ hyperplanes per cluster (instead of 1) that need to pass within distance $4\sqrt{\delta \log(t)/(d-1)}$ of the cluster center. This is because separating $m$ points needs $\log_2 m$ hyperplanes. Substituting this in the inequality (8) gives the bound in Theorem 5. $\qquad\square$

## C.1 Proof of Lemma 6

*Proof.* Let $\mathcal{P}_\delta$ be a $\delta$ packing of the unit sphere $\mathcal{S}^{d-1}$. Let $\mathcal{B}^d$ denote the unit ball, *i.e.,* the sphere $\mathcal{S}^{d-1}$ along with its interior. Then, if we draw balls of radius $\delta$ around each point in the packing, they will not intersect and all of them will be completely contained inside the ball of radius $1 + \delta$. Thus, we get the following

$$|\mathcal{P}_\delta| \times \text{Vol}(\delta \mathcal{B}^d) \le \text{Vol}((1 + \delta)\mathcal{B}^d)$$
$$\implies |\mathcal{P}_\delta| \le \frac{\text{Vol}((1 + \delta)\mathcal{B}^d)}{\text{Vol}(\delta \mathcal{B}^d)}$$
$$= \left(\frac{1 + \delta}{\delta}\right)^d$$
$$\le \left(\frac{2}{\delta}\right)^d.$$

Next, let $\mathcal{C}_\delta$ be a $\delta$-covering of the sphere $\mathcal{S}^{d-1}$. Any point on the sphere $(1 - \delta)\mathcal{S}^{d-1}$ is at distance $\delta$ of some point on $\mathcal{S}^{d-1}$. That point is, in turn, at distance within $\delta$ of some point in $\mathcal{C}_\delta$. Hence, by the triangle inequality, every point on on the sphere $(1 - \delta)\mathcal{S}^{d-1}$ is within distance $2\delta$ of some point in $\mathcal{C}_\delta$. Similarly, it can be shown that every point on on the sphere $(1 + \delta)\mathcal{S}^{d-1}$ is within distance $2\delta$ of some point in $\mathcal{C}_\delta$. Draw balls of radius $2\delta$ around each point in the covering $\mathcal{C}_\delta$. Then, these balls cover all the points within distance $\delta$ of the sphere $\mathcal{S}^{d-1}$. Thus, we get the following

$$|\mathcal{C}_\delta| \times \text{Vol}(2\delta \mathcal{B}^d) \ge \text{Vol}(((1 + \delta)\mathcal{B}^d) \setminus ((1 - \delta)\mathcal{B}^d)),$$
$$\text{which implies} |\mathcal{C}_\delta| \ge \frac{\text{Vol}((1 + \delta)\mathcal{B}^d) - \text{Vol}((1 - \delta)\mathcal{B}^d)}{\text{Vol}(2\delta \mathcal{B}^d)}$$
$$= \frac{(1 + \delta)^d - (1 - \delta)^d}{(2\delta)^d}$$
$$\ge \frac{(1 + \delta)^d - 1}{(2\delta)^d}$$
$$\ge \frac{1 + d\delta - 1}{(2\delta)^d}$$
$$= \frac{d\delta}{(2\delta)^d}$$
$$\ge \frac{1}{(4\delta)(d - 1)}.$$

Combining with the following inequality (Vershynin, 2018, Lemma 4.2.8):

$$\mathcal{C}_\delta \le \mathcal{P}_\delta,$$

we get

$$\left(\frac{1}{4\delta}\right)^{d-1} \le \mathcal{C}_\delta \le \mathcal{P}_\delta \le \left(\frac{2}{\delta}\right)^d.$$

$\square$

## D  Proofs from Section 6

### D.1  Proof of Theorem 3

| $\mathcal{H}\downarrow,\mathcal{P}\rightarrow$ | $p_1$ | $p_2$ | $p_3$ | $\ldots$ | $p_{|\mathcal{P}|}$ |
|---|---|---|---|---|---|
| $h_1$ | 1 | 0 | 0 | | 0 |
| $h_2$ | 1 | 1 | 1 | | 1 |
| $h_3$ | 0 | 1 | 0 | | 1 |
| $\vdots$ | | | | | |
| $h_{|\mathcal{H}|}$ | 0 | 1 | 0 | | 1 |

Table 2: (Lower bound) Here, if there are $r$ rows and $c$ columns, and $2^r < c$, then there will be two columns with the same pattern of 0's and 1's. Noting that $r = |\mathcal{H}|$ and $c = |\mathcal{P}|$ gives us the lower bound.

*Proof.* Let $\mathcal{A}$ output models from the hypothesis set $\mathcal{H}$. Then, we will essentially try to lower bound $\log_2 |\mathcal{H}|$, since we will need at least $\log_2 |\mathcal{H}|$ bits to represent the models.

The first lower bound is easy to see: For a fixed set of $n$ points, there can be $2^n$ possible assignments of labels. Hence we need at least $2^n$ hypothesis in our hypothesis class. Hence we need at least $n$ bits to memorize our dataset.

A lower bound with dependence on $\delta$ is also not difficult to get. Here is how we create our dataset: Create a $\delta$-packing $\mathcal{P}_\delta$ of the set $\mathcal{S}$. Choose any $n$ points from this packing. In fact, we will only concentrate on the first and the second point in our dataset. Assume that we have a hypothesis set $\mathcal{H}$ such that this can memorize these two points. This means that no matter which two points we choose from the packing and what labels we assign them, there will be a hypothesis in $\mathcal{H}$ that gives the chosen points the assigned labels. Now, if $2^{|\mathcal{H}|} < |\mathcal{P}_\delta|$, then there exist at least one pair of points in the packing $\mathcal{P}_\delta$ such that every hypothesis assigns the two points the same labels. To see how, consider Table 2. There can be at most $2^{|\mathcal{H}|}$ unique columns in that table. Hence, if $2^{|\mathcal{H}|} < |\mathcal{P}_\delta|$, then there would exist two points which have the same columns. We choose those two points and give them opposite labels, then this would contradict memorization, since all the hypotheses in $\mathcal{H}$ give them both the same label. This proves that $2^{|\mathcal{H}|} \geq |\mathcal{P}_\delta|$, or $\log_2 |\mathcal{H}| \geq \log_2 \log_2 |\mathcal{P}_\delta|$. Combined with the first lower bound, we get that we need at least $\max(n, \log_2 \log_2 |\mathcal{P}_\delta|)$ bits. $\square$

### D.2  Proof of Theorem 4

| $\mathcal{H}\downarrow,\mathcal{C}\rightarrow$ | $c_1$ | $c_2$ | $c_3$ | $\ldots$ | $c_{|\mathcal{C}|}$ |
|---|---|---|---|---|---|
| $h_1$ | 1 | 0 | 0 | | 0 |
| $h_2$ | 1 | 1 | 1 | | 1 |
| $h_3$ | 0 | 1 | 0 | | 1 |
| $\vdots$ | | | | | |
| $h_{|\mathcal{H}|}$ | 0 | 1 | 0 | | 1 |

Table 3: (Upper bound) Each entry of this table is i.i.d. Bernoulli with probability 0.5. All we need to do is to ensure that if we choose any $n$ columns above and any labeling in $\{0, 1\}^n$, there exists one row that has that same labeling.

*Proof.* Create a $\delta/2$ covering $\mathcal{C}_{\delta/2}$ of the set $\mathcal{S}$. We can partition the sphere into $|\mathcal{C}_{\delta/2}|$ regions of diameter less than $\delta$: Any point that lies within distance $\delta/2$ to any point in $\mathcal{C}_{\delta/2}$ is included in the region of that point. If a point lies within distance $\delta/2$ of multiple points in $\mathcal{C}_{\delta/2}$, then assign it to any one of the regions arbitrarily. Since by assumption, no two points in our dataset are within distance $\delta$, the benefit of this partition is that every pair of points in our dataset will lie in different

regions of the sphere. Consider the set of all the hypotheses $h_i$ that assign the same label to all the points which lie in a region of the partition. (If $\mathcal{S}$ is a sphere, then these hypotheses look like a football with black and white regions: black region is 0, white is 1). There will be $2^{\mathcal{C}_{\delta/2}}$ such hypotheses. We will only select a random subset of this large set and denote it by $\mathcal{H}$. We will show that $\log_2 \mathcal{H} = \mathcal{O}(n + \log_2 \log_2 |\mathcal{C}_{\delta/2}|)$ will suffice for memorization.

For any particular choice of $n$ points and $n$ labels, the probability that none of the $|\mathcal{H}|$ hypothesis have that choice of labels is $(1 - 2^{-n})^{|\mathcal{H}|}$. Taking a union bound over all the $\binom{|\mathcal{C}_{\delta/2}|}{n}$ choices of points and $2^n$ choices of labels, we need

$$\binom{|\mathcal{C}_{\delta/2}|}{n} 2^n (1 - 2^{-n})^{|\mathcal{H}|} < 1,$$

to ensure that there exists one $\mathcal{H}$ for which we can memorize any $n$ points. Noting that $1 - 2^{-n} < e^{-2^{-n}}$ and $\binom{|\mathcal{C}_{\delta/2}|}{n} < |\mathcal{C}_{\delta/2}|^n$, we see that it is sufficient to have

$$(2|\mathcal{C}_{\delta/2}|)^n e^{-2^{-n}|\mathcal{H}|} < 1.$$

Solving this, we get that $\log_2 |\mathcal{H}| > n + \log_2 n + \log_2 \log_e 2|\mathcal{C}_{\delta/2}|$ is sufficient, *i.e.*, we need $O(n + \log_2 \log_2 |\mathcal{C}_{\delta/2}|)$ bits to memorize $n$ points. $\qquad\square$