# OpenReview forum: "An Exponential Improvement on the Memorization Capacity of Deep Threshold Networks"
_NeurIPS.cc/2021/Conference — NeurIPS 2021 Poster_

### Official Review · Reviewer_JEGY · 2021-06-30

**Rating:** 8
**Confidence:** 4

**Summary:**

This paper considers the memorization of data using deep neural networks and aims to minimize both the number of neurons and the number of connections in the network, where the activation function is a step function.
The main result is a construction of a deep and relatively sparse network that memorizes arbitrary labels of 'well-separated' data sets.
If the size of the data set is $n$, the number of neurons in the network is proportional to $\sqrt{n}$ plus a linear dependence on the well-separateness parameter.
This is an exponential improvement over a recent result of Vershynin (which itself generalized a classical result of Baum from the 80's), which had an exponential dependence on the well-separateness parameter.
The authors also complement their work by proving some lower bounds, which, in particular, show that some polynomial dependence on the well-separateness parameter is necessary.

**Limitations And Societal Impact:**

The authors adequately addressed the limitations and potential negative societal impact of their work

**Main Review:**

The paper considers a well-established line of work concerning the memorization capacity of neural networks. The case of deep networks is particularly relevant and interesting. I find the construction introduced in the paper to be both intuitive and non-trivial.
It goes through a well-thought compression scheme followed by a separation procedure.
The compression allows to keep the network size small, and the separation procedure allows to memorize arbitrary labels.
The ideas are rather elegant and might even be of further use in other scenarios.

Moreover, the paper is particularly well written and does a pretty good job at introducing the relevant ideas and motivations.

Given the above, I am confident that this paper will be of interest to the NeurIPS community and am happy to recommend acceptance.

I would still like to raise a few questions/suggestions which I hope the authors can respond to:

- Line 22: This is completely nitpicky and is actually a matter of opinion, but I would say that the question of memorization can be tracked down to the 65' paper of Cover (in which he basically considers a network with a single neuron).

- Line 46: "...the $\delta$ and $n$ terms appear in summation rather than product...". I was slightly confused by this sentence since the same is also true for the bound proved by Vershnyn for the number of neurons. So maybe it would be better to say your specifically referring to the bound on the number of weights.

- The construction provided in the paper uses a deep network whose depth depends both on $n$ and on $\delta$. This is in contrast to Vershynin, which, as I remember, can work with a constant depth. Can the authors comment on this difference? Specifically, do they think an adaptive depth is necessary to achieve linear (or even polynomial) dependence on $\delta$?

- Line 93: I think that the comparison between Baum's and Vershynin construction is slightly misleading. It's true that having the points lie on a low dimensional hyperplane will violate the assumptions of Baum's construction, but as the authors note there is a trivial adaptation that will suffer some form of a 'dimensionality curse'. But this is also true for Vershynin construction. If the points lie on a low-dimensional hyperplane then $\delta$ should be polynomial in $\frac{1}{n}$ which will make for a network with a super polynomial number of neurons.

- Line 274: At first glance, taking the groups so that their coordinates do not overlap seems overly excessive to me. If we just wish the vectors to be linearly independent then surely some overlap can be tolerated.  I guess that the main risk is having very sparse vectors which may induce a linear dependence on the overlapping part. Is this the reason why having a substantial overlap will not work?

- The authors choose not to discuss generalization of neural networks since they claim their results cannot explain anything about generalization. I agree with this assessment, but I think a discussion about the resulting functions expressible by the authors' construction could be interesting. For example, we will certainly not expect any training algorithm like gradient descent to produce a similar network.
Another interesting point would be to compare it to the previous constructions. E.g. the construction of Baum results in a function that is essentially $0$ on the entire sphere, while Vershynin's construction will produce a random-looking function.

- Concerning Theorem 5. I was slightly disappointed by the fact that $n$ must depend on $\delta$. Of course, this is a necessity. otherwise, since the Theorem revolves around two-layered networks, it would contradict Baum's result.
However, the Theorem applies when $n \geq \frac{1}{\delta}$. We can plug this to main bound to get that the number of neurons must be at least $\sqrt{n}$, which is not new. In particular, it does not show that a dependence on $\delta$ is necessary for deep networks if one only wishes to use $\sqrt{n}$ neurons. Again, I would be happy if the authors can comment on that and whether they think any dependence on $\delta$ is necessary in this scenario.




**Time Spent Reviewing:**

8

---

> ### Author Response · Authors · 2021-08-10
> **Author response for Reviewer JEGY**
>
> We thank the reviewer for their time and effort. Please find our response to individual concerns below.
>
> ## Line 22: This is completely nitpicky and is actually a matter of opinion, but I would say that the question of memorization can be tracked down to the 65' paper of Cover (in which he basically considers a network with a single neuron).
>
> **Response:** Thank you for pointing this out, we will add this reference.
> ***
> ## Line 46: "...the $n$ and $\delta$ terms appear in summation rather than product...". I was slightly confused by this sentence since the same is also true for the bound proved by Vershynin for the number of neurons. So maybe it would be better to say your specifically referring to the bound on the number of weights.
>
> **Response:** Thank you for the suggestion, we will clarify this in the next version of the paper.
> ***
> ## The construction provided in the paper uses a deep network whose depth depends both on $n$ and on $\delta$. This is in contrast to Vershynin, which, as I remember, can work with a constant depth. Can the authors comment on this difference? Specifically, do they think an adaptive depth is necessary to achieve linear (or even polynomial) dependence on $\delta$?
>
> **Response:** Our network can be easily made as shallow as 4-hidden layers, while still maintaining a linear dependence on $\delta$. However, doing this will increase the number of neurons from $\widetilde{O}(\frac{1}{\delta}+\sqrt{n})$ to $\widetilde{O}(\frac{1+\sqrt{n}}{\delta})$ and the number of weights from $\widetilde{O}(\frac{d}{\delta}+n)$ to $\widetilde{O}(\frac{d+n}{\delta})$. To do this, we can simply remove the sparse subnetwork that compresses the binary representations from $\widetilde{O}(\frac{\log n}{\delta})$ to $\widetilde{O}(\log n)$ length binary vectors.
>
> Note that our current construction has depth $O(\log \frac{\log n}{\delta})$, and hence the dependence is logarithmic on $\delta$ and log log on $n$.
>
> ## Line 93: I think that the comparison between Baum's and Vershynin construction is slightly misleading. It's true that having the points lie on a low dimensional hyperplane will violate the assumptions of Baum's construction, but as the authors note there is a trivial adaptation that will suffer some form of a 'dimensionality curse'. But this is also true for Vershynin construction. If the points lie on a low-dimensional hyperplane then $\delta$ should be polynomial in $\frac{1}{n}$ which will make for a network with a super polynomial number of neurons.
>
> **Response:** Thank you for pointing this out, we should have explained the comparison in more detail. Consider the case when $d\gg \log n$. Note that $n$ points can be arranged in a $O(\log n)$ dimensional sphere while still ensuring that the distance between them is a constant, that is, $\delta$ is a constant. Hence, even though the ambient space is $d$ dimensional, the dataset lies on a $O(\log n)$ dimensional sphere. To visualize this, consider how in 3 dimensions, a dataset could lie on a circle (which is 2 dimensional). In this setting, Vershynin's bound would still need $\widetilde{O}(n+d)$ weights and $\widetilde{O}(\sqrt{n})$ neurons. On the other hand, naively extending Baum's construction to this setting would require $\widetilde{O}(nd)$ weights and $\widetilde{O}(n)$ neurons. We will add this detailed explanation in the paper.
> ***
> ## Line 274: At first glance, taking the groups so that their coordinates do not overlap seems overly excessive to me. If we just wish the vectors to be linearly independent then surely some overlap can be tolerated. I guess that the main risk is having very sparse vectors which may induce a linear dependence on the overlapping part. Is this the reason why having a substantial overlap will not work?
>
> **Response:** Yes, that is exactly the main reason.
>
> Another benefit of ensuring that the coordinates do not overlap was that we could achieve memorization with only integer weights, whereas simply linear independence might have led to real weights.
>
> To see how, note that we want to find a $\mathbf{w}$ such that $ \mathbf{Z}^{(3)}\mathbf{w} =\mathbf{y}$, where $\mathbf{w}$ is the weight vector, $\mathbf{Z}^{(3)}$ is the intermediate binary representation  of the datapoints (the rows of this matrix are the vectors $\mathbf{z}_i$ from equation (1) in out paper), and $\mathbf{y}$ is the vector of labels. There will always exist a $\mathbf{w}$ that satisfies this as long as $\mathbf{Z}^{(3)}$ is full rank (that is, $\mathbf{z}_i$ are linearly independent). Then, one possible solution would be $\mathbf{w}=(\mathbf{Z}^{(3)})^{+}\mathbf{y}$, where $(\mathbf{Z}^{(3)})^{+}$ is the pseudo inverse. However, the non-overlapping sparsity pattern helped us find an explicit value of $\mathbf{w}$ which contained only integers.
> ***
> ## The authors choose not to discuss generalization of neural networks since they claim their results cannot explain anything about generalization. I agree with this assessment, but I think a discussion about the resulting functions expressible by the authors' construction could be interesting. For example, we will certainly not expect any training algorithm like gradient descent to produce a similar network. Another interesting point would be to compare it to the previous constructions. E.g. the construction of Baum results in a function that is essentially  on the entire sphere, while Vershynin's construction will produce a random-looking function.
>
> **Response:** Thank you for this valuable suggestion. We will try to explain what kind of functions our construction leads to. Our initial intuition is that it divides the space into convex polytopes, and in certain regimes our construction might work similar to a nearest neighbor classifier.
> ***
> ## Concerning Theorem 5. I was slightly disappointed by the fact that $n$ must depend on $\delta$. Of course, this is a necessity. otherwise, since the Theorem revolves around two-layered networks, it would contradict Baum's result. However, the Theorem applies when $n\geq \frac{1}{\delta}$. We can plug this to main bound to get that the number of neurons must be at least $\sqrt{n}$, which is not new. In particular, it does not show that a dependence on $\delta$ is necessary for deep networks if one only wishes to use $\sqrt{n}$ neurons. Again, I would be happy if the authors can comment on that and whether they think any dependence on $\delta$ is necessary in this scenario.
>
>
> **Response:** This is a really good question. The dependence on $\delta$ is necessary when $n\gg d^2$, as the following example shows.
>
> Consider the following setting: We have a two datasets $\mathcal{D}_1$ and $\mathcal{D}_2$, both containing $n$ points and $n\gg d^2$. The separation for $\mathcal{D}_1$ is $\delta_1\approx d^2/n$, whereas the separation for $\mathcal{D}_2$ is $\delta_2 \approx n^{-2/d}$. Hence, $\delta_1\ll \delta_2$. Then, Theorem 2 says that any network that memorizes $\mathcal{D}_1$ will need a much wider first layer than a network that memorizes $\mathcal{D}_2$, and in particular, the former would need $\Omega(\sqrt{n}/d)$ neurons in the first layer, whereas the latter would need only $O(n^{2/d})$ neurons in the first layer (using Theorem 1). Hence, $\delta$ can indeed impact the architecture of the network, and difficult datasets (datasets with smaller separation) will need wider first layers. Note that, as you said, the overall network might still need $O(\sqrt{n})$ neurons, but our theorem says that at least the first layer need not be $O(\sqrt{n})$ wide.
>
>
> Referring to the statement "...Since the Theorem revolves around two-layered network...", we would like to point out that the theorem actually applies to any threshold network, just that it only gives a lower bound for the width of the first layer.

---

> > ### Comment · Reviewer_JEGY · 2021-08-17
> > **Response to rebuttal**
> >
> > Thank you very much for the clarification.
> >
> > - Our network can be easily made as shallow as 4-hidden layers
> >
> > This seems like an interesting point. I think that mentioning it in the paper along with the needed modification would be a good idea.
> >
> > - Our initial intuition is that it divides the space into convex polytopes, and in certain regimes our construction might work similar to a nearest neighbor classifier.
> >
> > Same comment as above. Space permitting, this could make for a very interesting discussion in the paper.

---

> > > ### Author Response · Authors · 2021-08-18
> > > **Author response**
> > >
> > > Thank you for the quick response and the suggestions. Yes, we plan to add these two discussions to the paper, if the space permits.

---

### Official Review · Reviewer_1QCv · 2021-07-13

**Rating:** 7
**Confidence:** 3

**Summary:**

This paper considers the problem of memorizing n data examples in R^d using deep threshold networks. It is proved that, if the dataset is delta-separated (either in angle or in l2 distance), then for any labels, there exists a threshold network with O( (log(n)^2/delta) + sqrt{n}log(n)^2) neurons and O( (d+log(n))log(n)/delta + nlog(n)^2) weights that can memorize the dataset. This improves prior rates which either depend linearly on n or exponentially on 1/delta. Additionally, this paper also studies bit and neuron complexity, and particularly it is proved that there exists a delta-separated dataset with n examples which requires Omega(log(n)/sqrt{delta}) neurons to memorize.

**Limitations And Societal Impact:**

I think it would be nice to include more comparison with prior results.
(1) Baum's construction requires n/d neurons, which is larger than sqrt{n} only when n>d^2. Even though the data assumption is different, it is still interesting to discuss the case where n<d^2.
(2) On line 233, it is mentioned that Yun et al. (2019) used a similar strategy; what are the differences?

Here are some typos:
(1) On line 88, "Veshynin" should be "Vershynin".
(2) On line 261, it seems that s_i^{(2)} should have dimension K instead of d^{(2)}.

**Main Review:**

This paper studies a fundamental problem, the number of neurons needed to memorize a given training set. I think the paper is clean and well-written; I appreciate the explanation of the proof ideas. There are some interesting ideas: while it seems standard to use Gaussian random vectors on the first layer to separate the training examples, as far as I know, it is a novel idea to compress the binary representations using XOR (implemented by a threshold layer). The idea to partition the dataset into subsets of size O(sqrt{n}log(n)) and learn each of them using a single neuron, and the construction in the lower bound are also interesting.

Update: Thank you for your response. I decide to increase my score to 7.

**Time Spent Reviewing:**

8

---

> ### Author Response · Authors · 2021-08-10
> **Author response for Reviewer 1QCv**
>
> We thank the reviewer for their time and effort. Please find our response to individual concerns below.
>
> ## Baum's construction requires n/d neurons, which is larger than $\sqrt{n}$ only when $n>d^2$. Even though the data assumption is different, it is still interesting to discuss the case where $n<d^2$.
>
> **Response:** If the data is in General Position (Baum's assumption) and $n<d^2$, then indeed Baum's construction requires fewer neurons. However, if $n\geq d^2$ or the dataset (or a subset of it) lies on lower dimensional space, then our construction requires fewer neurons.
>
> As an example, consider the case when $d\gg \log n$. Note that $n$ points can be arranged in a $O(\log n)$ dimensional sphere while still ensuring that the distance between them is a constant, that is, $\delta$ is a constant. Hence, even though the ambient space is $d$ dimensional, the dataset lies on a $O(\log n)$ dimensional sphere. To visualize this, consider how in 3 dimensions, a dataset could lie on a circle (which is 2 dimensional). Hence, our bound would still need $\widetilde{O}(n+d)$ weights and $\widetilde{O}(\sqrt{n})$ neurons. On the other hand, naively extending Baum's construction to this setting would require $\widetilde{O}(nd)$ weights and $\widetilde{O}(n)$ neurons.
>
> We will add the discussion about the cases when $n\geq d^2$ and when $n\leq d^2$ to our paper.
> ***
> ## On line 233, it is mentioned that Yun et al. (2019) used a similar strategy; what are the differences?
>
> **Response:** The strategy used by Yun et al. (2019) is to ensure that the vectors $\mathbf{z_i}$ in Equation (1) are linearly independent. We also do the same, but we go one step further and ensure that $\mathbf{z_i}$ are not just linearly independent, their sparsity patterns are such that their coordinates do not overlap as well. This helped us find a vector $\textbf{w}$ which had integer values and satisfies Equation (1). Further, since Yun et al. (2019) used ReLU activation and we used threshold activation, the way their network creates $\mathbf{z_i}$ is very different from how our network creates $\mathbf{z_i}$.
>
> Moreover, the rest of the network construction for the two papers is also quite different.
> ***
> ## (Typos) (1) On line 88, "Veshynin" should be "Vershynin". (2) On line 261, it seems that $s_i^{(2)}$ should have dimension K instead of $d^{(2)}$.
>
> **Response:** Thank you for pointing this out, we will fix the typo.

---

### Official Review · Reviewer_Dgbz · 2021-07-15

**Rating:** 7
**Confidence:** 3

**Summary:**

This paper studies the memorization capacity of deep threshold networks, and provides a significant improvement over previous work. For $n$ points in the $d$-dimensional unit ball with at least $\delta$ separation between any two points, it is proved that a threshold network with $\tilde{O} (\frac1\delta + \sqrt n)$ neurons and $\tilde{O} (\frac d\delta + n)$ weights can memorize them. The previous best result (Vershynin, 2020) has an $e^{1/\delta^2}$ dependence in these bounds, and thus the current paper provides an exponential improvement. This paper also provides a lower bound saying that for $n$ in a certain range, the required number of neurons in the first layer must be $\tilde{\Omega}(1/\sqrt\delta)$, though this does not match the upper bound in the paper.

**Limitations And Societal Impact:**

The authors perhaps could discuss what should be the right dependence on $\delta$, since there is a discrepancy in the upper and lower bounds provided.

**Main Review:**

Originality: The contribution is original and provides a clear improvement on a theoretical problem studied in the past. The discussion of related work is sufficient.

Quality: The paper is technically sound. The theoretical results are clearly stated, and proof sketches are given in the main text.

Clarity: The paper is clearly written and well organized. The proof sketches are easy to follow.

Significance: Although the memorization power of neural networks has been studied for decades, the significance to the current practice is unclear, because the existence of a network to interpolate datapoints doesn't mean that such network can be found efficiently (almost always the theoretical constructions are different from the ones actually algorithmically found), let alone saying anything about generalization. In addition, the threshold activation is not practically adopted, to the best of my knowledge. That said, from a pure theoretical perspective, this paper studies an existing well-formulated question and obtains a strong improvement. I find the construction in this paper very neat.

Other comments:
- I find it a bit weird to define memorization in terms of a learning algorithm (Definition 1), since it's really just about the existence of a neural network.
- line 308: The packing/covering number should be $\Theta((C/\delta)^d)$ for some constant $C$. Though it doesn't matter after taking logarithm.

**Time Spent Reviewing:**

3

---

> ### Author Response · Authors · 2021-08-10
> **Author response for Reviewer Dgbz**
>
> We thank the reviewer for their time and effort. Please find our response to individual concerns below.
>
> ## Although the memorization power of neural networks has been studied for decades, the significance to the current practice is unclear, because the existence of a network to interpolate datapoints doesn't mean that such network can be found efficiently (almost always the theoretical constructions are different from the ones actually algorithmically found), let alone saying anything about generalization. In addition, the threshold activation is not practically adopted, to the best of my knowledge.
>
> **Response:** Our memorization result actually provides a constructive proof, that is, we explicitly show how each layer of the network can be constructed in randomized polynomial time for perfect memorization. We will add a note about this in the paper.
>
> Our work provides no insights on the learning capabilities of threshold networks, and we mention this explicitly in our paper. We are of the opinion that even before understanding generalization, the memorization capacity of a neural network is scientifically interesting, as evident by a long line of deep theoretical work on this subject.
>
> It is true that deep threshold networks are not widely used in practice, likely because they cannot be trained using the conventional gradient based algorithms. However, our paper shows that threshold networks that interpolate can be constructed efficiently in randomized poly-time, without gradient based algorithms.
> It is a very interesting future direction to develop training algorithms for deep threshold networks that lead to good generalization.
> This would be beneficial because threshold networks provide benefits such as faster inference, etc. These are important topics that fall beyond the scope of our work.
> ***
> ## I find it a bit weird to define memorization in terms of a learning algorithm (Definition 1), since it's really just about the existence of a neural network.
>
> **Response:** We considered a general definition of  memorization as one can potentially ask the same question regarding other type of models such as a nearest neighbour, decision trees, etc.
> ***
> ## The packing/covering number should be  for some constant . Though it doesn't matter after taking logarithm.
>
> **Response:** Thank you for pointing this out, we will change that to $\theta((C_p/\delta)^d)$ for packing number and
> $\theta((C_c/\delta)^d)$ for covering number for covering number. The reason we skipped the constants was because, as you said, the constants do not matter after taking logarithm.
>
> ***
> ## The authors perhaps could discuss what should be the right dependence on $\delta$, since there is a discrepancy in the upper and lower bounds provided.
>
> **Response:** This is a very important open question, and we do not have the answer for this right now.
>
> We think that the $O(\frac{1}{\delta})$ dependence on the number of neurons (Theorem 1) in the first layer is necessary, at least when $n=\Omega(d^2/\delta)$ (the setting of Theorem 2). In this case, the lower bound of Theorem 2 might actually be loose and indeed $\Omega(1/\delta)$ neurons might indeed be necessary. However, this is just speculation based on intuition, and more research will be needed to answer this open question.
>
> We will add this comment to the paper.

---

> > ### Comment · Reviewer_Dgbz · 2021-08-24
> > **Response from reviewer**
> >
> > Thanks to the authors for the response. I agree that this is a good paper and am keeping my score.

---

### Official Review · Reviewer_JGb5 · 2021-07-18

**Rating:** 7
**Confidence:** 3

**Summary:**

The paper considers a feed-forward neural network that takes inputs in $\mathbb{R}^d$ and outputs binary labels. The neural network has $L$ layers, with each layers consisting of a bias term and the activation function is the threshold function. The paper considers the following question: What is the minimum size of a threshold network so that it can memorize a dataset of $n$ samples in $\mathbb{R}^d$. Memorization refers to learning the parameters of the threshold network so that it can accurately output labels of the $n$ samples, for any arbitrary labeling.
The paper shows that a deep threshold network can memorize $n$ points in $d$ dimension using $O(\frac{1}{\delta} + \sqrt{n})$ neurons $O(\frac{d}{\delta} +n)$ weights (up to log factors), where $\delta$ is the minimum distance between the points (considered in angular metric and $\ell_2$-metric).

**Limitations And Societal Impact:**

The work does not have any immediate negative societal impact. Limitation of the work is not addressed with sufficiently. See main review.

**Main Review:**

The paper is well written and pleasant to read. Bulk of the paper rests on the theoretical exposition. The paper provides a good description of prior work. I did not go through the theoretical proofs from the supplementary material in detail, however the proof sketch from the
main paper provides sufficient intuition.

The work on the paper is incremental and provides an improved dependence on the $\delta$ for memorization using deep threshold networks. The improvement is from previously shown exponential dependence of $O(e^{1/\delta^2} + \sqrt{n})$ neurons and $O(e^{1/\delta^2}(d +\sqrt{n}) + n)$ weights to almost linear dependence on $\delta$ of $O(\frac{1}{\delta} + \sqrt{n})$ neurons $O(\frac{d}{\delta} +n)$ weights. Is the $O(\frac{1}{\delta})$  and $O(\frac{d}{\delta})$ dependence on $\delta$ fundamental or can it be improved further? A discussion on this would be helpful for the readers to gauge the importance of the result presented in the paper.

The work by [Veshynin, 2020] that showed the exponential dependence $\delta$ and $O(\sqrt{n})$ dependence on $n$ applies to both ReLU activation network and hard threshold activation network. Is the result presented in the current paper transferable to ReLU activation network? Is a similar dependence on $\delta$ expected there? A discussion on this would be useful as ReLU activation function are common and easy to implement in practice.

The paper constructs the threshold activation network with random weights on the first layer and subsequent layers have integer (binary) weights. Comments on whether integer restriction can be relaxed and still achieve similar performance on memorization would be useful.

The paper considers binary labeling. Comments on memorization problem with multiple labels (more than 2) would be useful as classification problems can have multiple labels.

The goal of learning a networks for classification is generally to use the learned network to correctly label points the network was not trained on. The paper does not address this issue but to state this in passing. A section detailing limitation of the work would be useful.


**Time Spent Reviewing:**

5

---

> ### Author Response · Authors · 2021-08-10
> **Author response for Reviewer JGb5**
>
> We thank the reviewer for their time and effort. Please find our response to individual concerns below.
>
> ## Is the $O(\frac{1}{\delta})$ and $O(\frac{d}{\delta})$ dependence on $\delta$ fundamental or can it be improved further?
> **Response:** This is a very important open question, and we do not have an exact answer for this right now.
> We think that the $O(\frac{1}{\delta})$ dependence on the number of neurons in the first layer is necessary, at least when $n=\Omega(d^2/\delta)$ (the setting of Theorem 2). In this case, the lower bound of Theorem 2 might actually be loose and indeed $\Omega(1/\delta)$ neurons might indeed be necessary.
>
> If $\Omega(\frac{1}{\delta})$ neurons are indeed necessary in the first layer, then the first layer's weight matrix will have dimensions $\Omega(\frac{1}{\delta})\times d$. Hence, naively counting the number of parameters gives us that $d\cdot\Omega(\frac{1}{\delta})=\Omega(\frac{d}{\delta})$ parameters are necessary. However, it is possible the first layer's matrix is constructed to be sparse, in which case the dependence could be improved.
> However, this is just speculation based on intuition, and more research will be needed to answer this open question.
> ***
>    ## Is the result presented in the current paper transferable to ReLU activation network? Is a similar dependence on  expected there?
> **Response:** Threshold activations can be approximated using two ReLU activations, so the results can be extended to ReLU networks. However, the norm of the weights can increase.
>
> Since ReLU networks are able to pass on the amplitude information to deeper layers, they might be more efficient at memorization. Indeed Park et al. (2020) show that ReLU networks can memorize with fewer neurons and parameters. However, our lower bounds from Section 6 will still apply to ReLU networks.
>     We will add a discussion about this in the paper.
> ***
> ## Comments on whether integer restriction can be relaxed and still achieve similar performance on memorization would be useful.
> **Response:** The construction doesn't necessarily need integer weights - there may exist other, non-integer real weights that can be used instead. However, the only reason for using (small) integer weights was because small integers can be stored efficiently, thus decreasing the model size in memory. Thus, integer weights aren't a restriction, but a beneficial feature of our construction.
> The reason we were able to use integer weights is that the outputs of each layer are binary vectors (due to threshold activation) instead of real vectors. Since integer weights are sufficient for manipulating binary vectors, we didn't need to use real weights.
>
> Whether the integer weights can further be completely replaced by just binary weights is an open question. We would like to point out that even in our construction, most of the weights after the first layer are binary and only the biases are integers.
> ***
> ## Comments on memorization problem with multiple labels (more than 2) would be useful as classification problems can have multiple labels.
> **Response:** The construction can be directly extended to multiple labels by modifying the last two layers. Multi-class classification with $m$ classes can be viewed as $m$ binary classification tasks, where the $i$-th task is to classify whether the sample belongs in the $i$-th class. We will add a detailed explanation in the appendix.
> ***
> ## The goal of learning a network for classification is generally to use the learned network to correctly label points the network was not trained on. The paper does not address this issue but to state this in passing.
> **Response:** Our work offers no clear insights or explanations with regards to why trained deep networks generalize well. We decided to omit any loosely made insights for the purpose of clarity and honesty. We will add comments clearly stating the scope and limitations of this work.

---

### Author Response · Authors · 2021-08-10
**Author response for the reviewers**

We thank all the reviewers for their time and effort in providing feedback. We also thank the reviewers for their valuable suggestions.

We are encouraged by the universally positive scores (8, 7, 7, 6) and that the reviewers appreciated the paper for providing a significant improvement on previous work (JGb5, Dgbz, 1QCv, JEGY), having novel and interesting ideas (1QCv, JEGY), being technically sound (Dgbz), being well written with clear explanation of the theory (JGb5, Dgbz, 1QCv, JEGY), and discussing the prior work sufficiently (JGb5, Dgbz).

We would like to briefly restate the results of this work. We prove that deep threshold networks can efficiently memorize datasets; and offer an exponential improvement to existing bounds. Vershynin (2020) recently proved that deep threshold network can memorize $n$ points in $d$ dimensions as long as the number of neurons is $\widetilde{O}(e^{1/\delta^2}+\sqrt{n})$ and the number of weights is $\widetilde{O}(e^{1/\delta^2}(d+\sqrt{n})+n)$, where $\delta$ is the separation between points. We improve the dependence on $\delta$ **exponentially**, bringing it down to almost linear: We show that actually only $\widetilde{O}(\frac{1}{\delta}+\sqrt{n})$ neurons and $\widetilde{O}(\frac{d}{\delta}+n)$ weights are needed.

We have provided individual responses to the Reviewer concerns as well.

---

### Decision · Program_Chairs · 2021-09-27

**Decision:**

Accept (Poster)

**Comment:**

A solid progress in a well studied line of research